# A morpheein equilibrium regulates catalysis in phosphoserine phosphatase SerB2 from *Mycobacterium tuberculosis*

Elise Pierson [1], Florian De Pol[1], Marianne Fillet[2] & Johan Wouters [1]✉

*Mycobacterium tuberculosis* phosphoserine phosphatase *Mt*SerB2 is of interest as a new antituberculosis target due to its essential metabolic role in L-serine biosynthesis and effector functions in infected cells. Previous works indicated that *Mt*SerB2 is regulated through an oligomeric transition induced by L-Ser that could serve as a basis for the design of selective allosteric inhibitors. However, the mechanism underlying this transition remains highly elusive due to the lack of experimental structural data. Here we describe a structural, biophysical, and enzymological characterisation of *Mt*SerB2 oligomerisation in the presence and absence of L-Ser. We show that *Mt*SerB2 coexists in dimeric, trimeric, and tetrameric forms of different activity levels interconverting through a conformationally flexible monomeric state, which is not observed in two near-identical mycobacterial orthologs. This morpheein behaviour exhibited by *Mt*SerB2 lays the foundation for future allosteric drug discovery and provides a starting point to the understanding of its peculiar multifunctional moonlighting properties.

[1] Laboratoire de Chimie Biologique Structurale (CBS), Namur Research Institute for Life Sciences (NARILIS), University of Namur (UNamur), 5000 Namur, Belgium. [2] Laboratory for the Analysis of Medicines (LAM), Center for Interdisciplinary Research on Medicines (CIRM), University of Liège (ULiège), 4000 Liège, Belgium. ✉email: johan.wouters@unamur.be

Recently overthrown by SARS-CoV2 from being the leading infectious killer worldwide, *Mycobacterium tuberculosis* (*Mtb*), the bacterial pathogen causing tuberculosis (TB), is at the origin of an estimated 1.6 million deaths in 2021. Although the current antibiotic treatment successfully cures 85% of the infected patients, its costly and fastidious nature often results in non-adherence or mismanagement contributing to the high prevalence of the disease[1]. In addition to the COVID-19 pandemic reversing years of progress in the global strategy against TB[2], the emergence and spread of multidrug-resistant *Mtb* strains further complicates the global TB problem and urge the need to extend our therapeutic weaponry[3]. In this context, the essential enzymes of *Mtb* amino acid metabolism have gained strong interest and could be the targets of an upcoming generation of antitubercular drugs[4–7].

*Mt*SerB2 is a 43 kDa phosphoserine phosphatase (PSP) that catalyses the dephosphorylation of *O*-phospho-L-Ser to L-serine (L-Ser), the last irreversible step of the L-Ser biosynthetic pathway in *Mtb*. In 2003, Sassetti et al. highlighted that the gene encoding *Mt*SerB2 is critical for the optimal growth of *Mtb* in vitro[8]. This pioneering work galvanised further studies focusing on the enzyme. It was later discovered that its phosphatase activity could also be related to non-metabolic functions contributing to the intra-macrophage survival of *Mtb* and its escape from the host immune response[9,10].

Inhibiting *Mt*SerB2 has therefore been regarded with interest to identify antitubercular agents with a novel mechanism of action[11–13]. However, a thorough structural characterisation of *Mt*SerB2 is still lacking, which hampers structure-based drug design strategies. To date, observations have been rationalised using a homology model of *Mt*SerB2 based on the crystallographic structure of a close orthologous PSP from *Mycobacterium avium*[14] (*Ma*SerB, 83.7% sequence identity with *Mt*SerB2). According to homology modelling, *Ma*SerB and *Mt*SerB2 share a unique and identical domain-swapped homodimeric architecture. *Mt*SerB2 is thus predicted as two intertwined subunits related by C2 symmetry (Fig. 1a), each bearing a C-terminal catalytic PSP domain composed of a haloacid dehalogenase Rossmannoid core capped with a C1 type module[15], and two consecutive regulatory ACT domains linked together by a 12 residues-long flexible hinge-loop in N-terminus (Fig. 1b). The extension of the hinge-loop allows the formation of the intertwined butterfly-like homodimer through the inter-subunit exchange of the N-terminal ACT1 domain and its interaction with the ACT2 domain to create an eight-stranded antiparallel beta-sheet with four helices on one side (Fig. 1c). This dimeric arrangement of ACT domains has been observed in other enzymes, such as *E. coli* phosphoglycerate dehydrogenase (PGDH, Fig. 1d), and allows the regulation of catalysis from allosteric signals transmitted through the binding of specific ligands at the dimeric interface[16–18].

Interestingly, Yadav et al. showed that *Mt*SerB2 homodimer undergoes an oligomeric transition to an inactive higher-order oligomer in the presence of L-Ser[9]. This transition is a supposed mechanism for the endogenous feedback inhibition of the enzyme, which is coherent with Grant's kinetic studies indicating that L-Ser allosterically regulates *Mt*SerB2 catalytic activity[19]. Although these works both suggest that L-Ser interacts with the ACT domains, the structural mechanism underlying the formation of the regulatory oligomer remains elusive. We decided to further investigate this mechanism.

Here, we present the results that allowed us to dissect the intricate complexity of *Mt*SerB2 oligomeric behaviour. Through synergistic biophysical and biochemical experiments performed in the absence and presence of L-Ser, we demonstrate the structural, catalytic, and interconversion properties of three distinct quaternary states of *Mt*SerB2. We show how our findings converge to the conclusion that *Mt*SerB2 possesses unique morpheein properties that distinguish it from two near-identical mycobacterial orthologs sharing over 83% sequence identity. While the

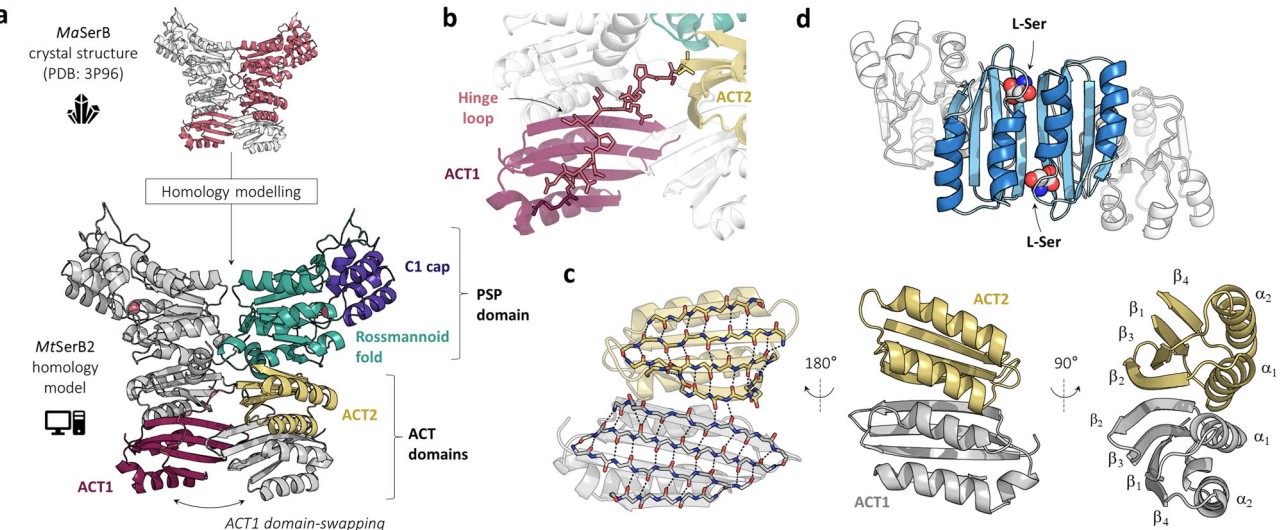

**Fig. 1 Overview of *Mt*SerB2 structural features based on homology modelling. a** *Mt*SerB2 homology model (bottom) constructed on the basis of *Ma*SerB crystal structure[14] (top, 83.7% sequence identity). The two predicted monomer chains are respectively depicted in grey and in colour. Structural regions of interest are highlighted: C-terminal C1 cap module (purple), Rossmannoid fold (green), N-terminal ACT domains (yellow and red). The predicted magnesium ions cofactors are represented as pink spheres inside the Rossmannoid fold of each monomer chain. The two monomer chains are related by C2 symmetry and exchange their ACT1 domain to form a dimer with the ACT2 domain of the other monomer chain (**c**). **b** Predicted conformation of the 12 residues-long hinge region separating ACT2 and ACT1 domains, and allowing ACT1 domain-swapping in *Mt*SerB2 homology model. **c** Intermolecular dimeric arrangement of ACT1 and ACT2 domains in *Mt*SerB2 homology model. The ACT dimer exhibits an eight-stranded antiparallel beta-sheet in the order 4–1–3–2–2–3–1–4 on one side and four helices in the order 2–1–1–2 on the other. One monomer chain is depicted in grey and the other in yellow. **d** The archetypical dimeric ACT domain arrangement of *E. coli* phosphoglycerate dehydrogenase tetramer[85] (PDB: 1PSD) with two L-Ser molecules bound at the interface.

existence of such quaternary dynamics might play a role in the virulence factor properties observed in *Mt*SerB2, it could also create new avenues to achieve selective toxicity towards *Mtb* and avoid collateral inhibition of the human PSP.

## Results

**MtSerB2 mainly exists as slowly interconverting active dimeric and inactive tetrameric states in serine-free solution.** We began by investigating *Mt*SerB2 oligomerisation in buffer conditions similar to those already reported in the literature[9–11]. The use of analytical size-exclusion chromatography coupled with ultraviolet and multi-angle light scattering (SEC-UV-MALS) detectors highlighted that *Mt*SerB2 (43 kDa) exists as three distinct oligomeric populations under these conditions (Fig. 2a): a majority dimer (87 kDa, 68.6% of area under the UV curve (AUC)), a minority tetramer (165 kDa, 26.7% AUC) and a trace population whose molar mass approaches that of a monomer (56 kDa, 4.7% AUC). The ratios were reproducible between samples from different purification batches and thus should reflect population concentrations at equilibrium. Additional SEC-UV-MALS experiments showed that the two close orthologs *Ma*SerB (83.3% sequence identity) and *Mm*SerB2 (85.7% sequence identity) exist almost exclusively in a dimeric state under the same conditions (Fig. 2b, c). The associated chromatograms also indicate the presence of a trace population that may be monomeric, but no tetrameric species were observed in these samples.

The interconversion rate between *Mt*SerB2 dimeric and tetrameric populations was investigated. When analysing their composition by native PAGE after preparative SEC, we systematically observed that the tetramer peak fraction pool contained a tetrameric population contaminated by a small amount of dimer, and that the dimer peak fraction pool only contained pure dimer (Fig. 2d, lanes 1 and 3). No change in oligomeric equilibrium was observed when the pools were concentrated to a high total protein concentration and directly analysed by electrophoresis (Fig. 2d, lanes 2 and 4). Flash-frozen enriched dimer and tetramer fraction pools were also analysed by mass photometry (MP) directly after thawing and at time points of two weeks and three months after thawing (Fig. 2e). We found that the tetramer-to-dimer conversion was faster than the dimer-to-tetramer conversion. After three months, dimer/tetramer ratios approached the 85/15 equilibrium ratio measured in the unseparated sample during the same MP analysis series (Supplementary Fig. 1). All in all, these results suggest that *Mt*SerB2 dimer is more stable than *Mt*SerB2 tetramer, and that their measurable interconversion could be governed by a more subtle mechanism than a straightforward oligomeric aggregation/disaggregation phenomenon.

Finally, we assessed the respective activities of *Mt*SerB2 dimer and tetramer after their separation by preparative SEC. The purity of each population fraction pool was first verified by native PAGE (Fig. 3a). Then, steady-state kinetics experiments were performed by quantifying the phosphate released during the dephosphorylation of phosphoserine at varying concentrations by each population fraction pool. The single band on the gel (Fig. 3a, lane 1) demonstrated the purity of the dimeric pool, and the related plot of the dimeric pool activity versus phosphoserine concentration could be fitted to the standard equation for uncompetitive substrate inhibition (Eq. (1) of 'Methods'), with a catalytic constant ($k_{cat}$) of 39.9 ± 7.8 s$^{-1}$, a Michaelis constant ($K_M$) of 0.48 ± 0.17 mM, and a substrate inhibition constant ($K_{iS}$) of 2.26 ± 0.70 mM (Fig. 3b, c). Although a higher $K_{iS}$ value was reported by Grant[19], our results are in agreement with the kinetic behaviour concluded by the author. Similar kinetic behaviours were also observed for *Ma*SerB and *Mm*SerB2 dimers (Supplementary Fig. 2). According to MP measurements, *Mt*SerB2

tetrameric pool contained 45.9% of the dimer (Fig. 3d). We suspected that only the dimer is an active species. By substituting the total protein amount by the amount of dimer alone in the calculation of the kinetic parameters of the tetramer fraction pool (Fig. 3e), the obtained $k_{cat}$ value (39.4 ± 5.7 s$^{-1}$) was equivalent to that of the pure dimer fraction pool (39.9 ± 7.8 s$^{-1}$). This indicates that the tetrameric population indeed does not dephosphorylate phosphoserine, nor binds it since the $K_M$ value did not vary (0.48 ± 0.17 mM vs. 0.43 ± 0.12 mM).

**MtSerB2 dimer exhibits an ACT1 domain-swapped butterfly-like architecture similar to MaSerB.** We then performed SEC-UV-SAXS analyses on both *Mt*SerB2 dimer and its close ortholog *Ma*SerB to experimentally investigate *Mt*SerB2 dimer architecture in solution. The scattering curves were obtained from monodisperse elution peaks (Supplementary Fig. 3) and overlap well (Fig. 4a), as do the $P(r)$ functions calculated by indirect Fourier transform (Fig. 4b). The $R_g$ values determined by Guinier analysis and from the $P(r)$ function, as well as the determined maximal protein dimensions $D_{max}$ and Porod volumes $V_p$ are very similar for both enzymes (Table 1). The dimensionless Kratky plots match each other and their bell shape exhibiting a maximum value of 1.104 for $qR_g = \sqrt{3}$ indicate that the orthologs are globular well-folded species (Fig. 4c). The overall comparison highlights identical SAXS behaviour for both proteins and therefore strongly suggests that *Mt*SerB2 dimer is folded similarly to *Ma*SerB in solution. The same analysis workflow allowed us to draw the same conclusion regarding *Mm*SerB2 architecture (Table 1 and Supplementary Fig. 4). Additionally, we computed the theoretical SAXS curve for the crystallographic structure of *Ma*SerB[14] to evaluate the variation from the solution structure. The superposition of the predicted curve to the experimental data resulted in an unsatisfying fit with systematic deviation from experimental data from q = 0.125 Å$^{-1}$ (Fig. 4d). We consequently performed a SAXS-guided structural relaxation on *Ma*SerB crystal structure[14] to know whether slight structural changes could result in a better fit. Excellent fits were obtained after the relaxation, and the obtained models, although showing slight differences in the position of the ACT1 domain, retained the domain-swapped scaffold.

However, the resolution of structural information provided by SAXS is not sufficient to confirm domain-swapping in *Mt*SerB2. According to domain-swapping principles, a hinge-loop motion is responsible for the opening of a closed *Mt*SerB2 monomer, which further associates with another opened monomer to form the domain-swapped dimer. The opening of the closed monomer is a repositioning of the ACT1 domain and does not involve any refolding. The ACT1-ACT2 intramolecular interactions that are broken upon monomer opening are replaced by identical ACT1-ACT2 intermolecular interactions in the dimer. This way, the relative positions of the folded PSP, ACT2, and ACT1 domains in the domain-swapped dimer is strictly identical to the relative positions of these domains in the closed monomer (Supplementary Fig. 5). Hence, a non-domain-swapped *Mt*SerB2 dimer where two closed monomers interact gives a SAXS signature similar to that of the domain-swapped dimer, because the positions of the domains relative to each other are identical and the overall molecular shape is conserved. Therefore, an interface analysis was performed to highlight the importance of forming the intermolecular ACT1-ACT2 interface for *Mt*SerB2 dimerisation and to support dimerisation by ACT1 domain-swapping.

*Mt*SerB2 domain-swapped dimer homology model and a non-domain-swapped dimer model created in silico from two closed monomers models were submitted to PDBePISA web server. The interface area calculated for the domain-swapped dimer

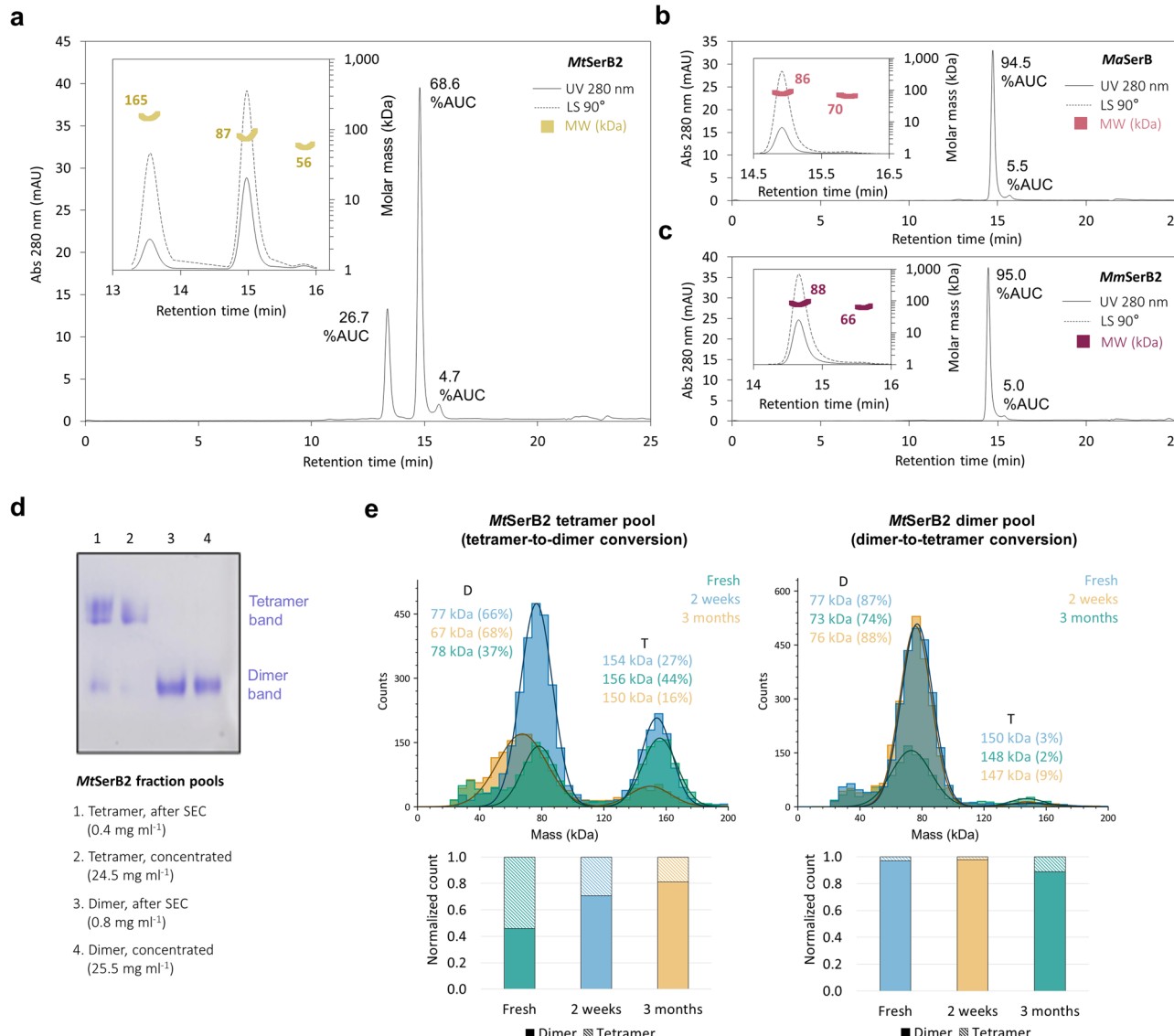

**Fig. 2 Identification of *Mt*SerB2 dimeric and tetrameric populations by SEC-UV-MALS and study of their interconversion by native PAGE and mass photometry (MP). a** SEC-UV-MALS analysis of *Mt*SerB2 (43 kDa). The main chromatogram shows the elution monitored by absorbance at 280 nm and the inset gives a close-up on this signal, superimposed on the light scattering signal measured at a 90° angle. In elution order, the peaks correspond to: tetramer (expected molar mass 172 kDa), dimer (expected molar mass 86 kDa), and monomer, probably in equilibrium with higher-order oligomers as indicated by the higher measured molar mass. **b** SEC-UV-MALS analysis of *Ma*SerB (44 kDa). In elution order, the peaks correspond to: dimer (expected molar mass 88 kDa), and monomer, probably in equilibrium with higher-order oligomers as indicated by the higher measured molar mass. **c** SEC-UV-MALS analysis of *Mm*SerB2 (44 kDa). In elution order, the peaks correspond to: dimer (expected molar mass 88 kDa), and monomer probably in equilibrium with higher-order oligomers as indicated by the higher measured molar mass. **d** Native PAGE analysis of *Mt*SerB2 after preparative SEC. Tetramer peak fraction pool (lanes 1 and 2) and dimer peak fraction pool (lanes 3 and 4) directly after eluting from the SEC column and directly after concentration by centrifugation. The tetramer fraction pool is composed of a majority tetrameric population (upper band), contaminated by a dimeric population (lower band), whereas the dimer fraction pool only contains a dimeric population. The similar band pattern observed before and after concentration shows that concentration does not affect the oligomeric equilibrium instantaneously. **e** Mass photometry histograms of *Mt*SerB2 tetramer peak fraction pool and dimer peak fraction pool. Both pools were flash-frozen after SEC separation and analysed at different time points after thawing: directly after thawing, after 2 weeks at 4 °C and after 3 months at 4 °C. The percentage of total counts is indicated in brackets. The bottom histograms show the change in the ratio of the two populations contained in both pools as their normalised count for each time point. D dimer, T tetramer.

(3025.9 Å²) was about 2.5× larger than the one of the non-domain-swapped dimer (1222.3 Å²). The solvation-free energy gain ($\Delta^i G$) value computed for the formation of the interface in the domain-swapped model (−31.4 kcal/mol) was 4.4× the one calculated for the non-domain-swapped model (−7.1 kcal/mol). The ACT1-ACT2 intermolecular interface therefore accounts for 60% of the total dimeric interface and more than 75% of the total solvation-free energy gain. These values support that the

formation of the intermolecular ACT1-ACT2 interface by domain-swapping plays a significant role in stabilising *Mt*SerB2 dimer. Additionally, the monomeric behaviour of an engineered *Mt*SerB2 variant lacking the N-terminal ACT1 domain (*Mt*SerB2-ΔACT1) assessed by SEC-UV-MALS (Supplementary Fig. 6) further consolidates our findings by underlining the importance of the ACT1 domain for dimerisation and hence supporting the existence of ACT1 domain-swapping in *Mt*SerB2. From these

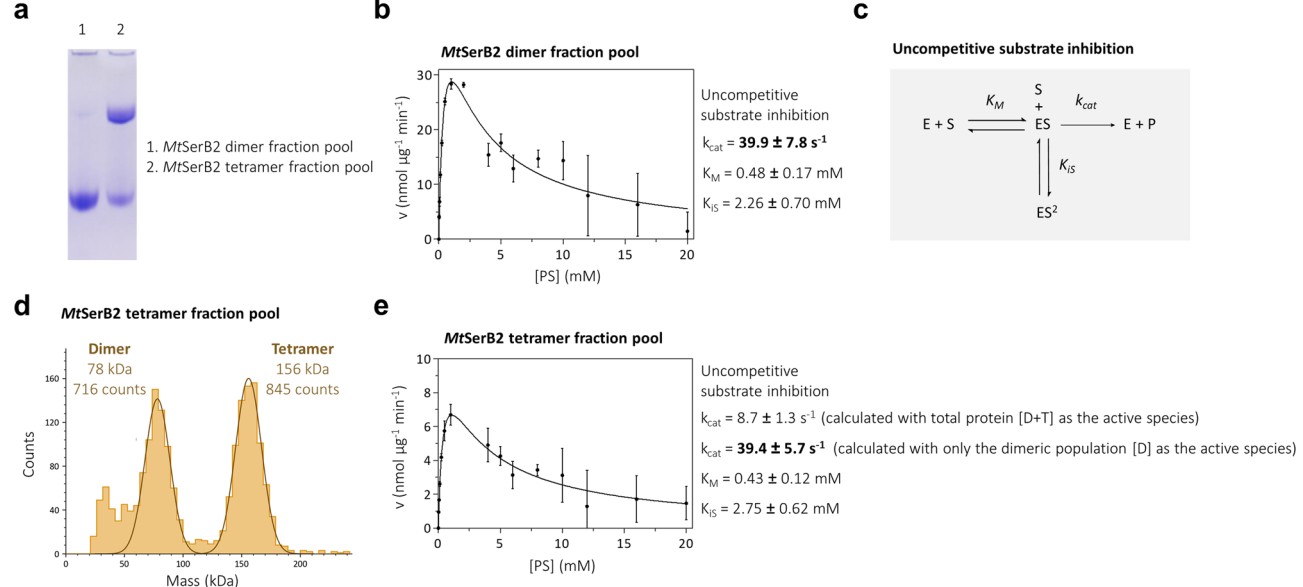

**Fig. 3 Evaluation of the kinetics of phosphoserine (PS) dephosphorylation by *Mt*SerB2 dimer and *Mt*SerB2 tetramer. a** Native PAGE analysis assessing the purity of *Mt*SerB2 dimer (lane 1) and tetramer (lane 2) peak fraction pools after separation by SEC. The single lower band in lane 1 indicates that the dimer fraction pool only contains a dimeric population, whereas the presence of two bands in lane 2 indicates that the tetramer fraction pool is composed of a majority tetrameric population (upper band), contaminated by a dimeric population (lower band). **b, e** Plots of initial velocity (nmol phosphate released per μg of *Mt*SerB2 per minute) versus substrate concentration for PS dephosphorylation by *Mt*SerB2 dimer fraction pool (**b**) and tetramer fraction pool (**e**). Error bars represent the s.d. of three experiments. The corresponding kinetic parameters were calculated by fitting Eq. (1) describing total uncompetitive substrate inhibition and are shown next to the plots. **c** Mass photometry distribution of *Mt*SerB2 tetramer peak fraction pool. According to the measured total counts (1561), the sample contains 45.9% dimer (716 counts) and 54.1% tetramer (845 counts). **d** Total uncompetitive substrate inhibition mechanism. E enzyme, S substrate (phosphoserine), P product (phosphate), $k_{cat}$ catalytic constant, $K_M$ Michaelis constant, $K_{iS}$ substrate inhibition constant. Source data are available on the FigShare repository (https://doi.org/10.6084/m9.figshare.24116571).

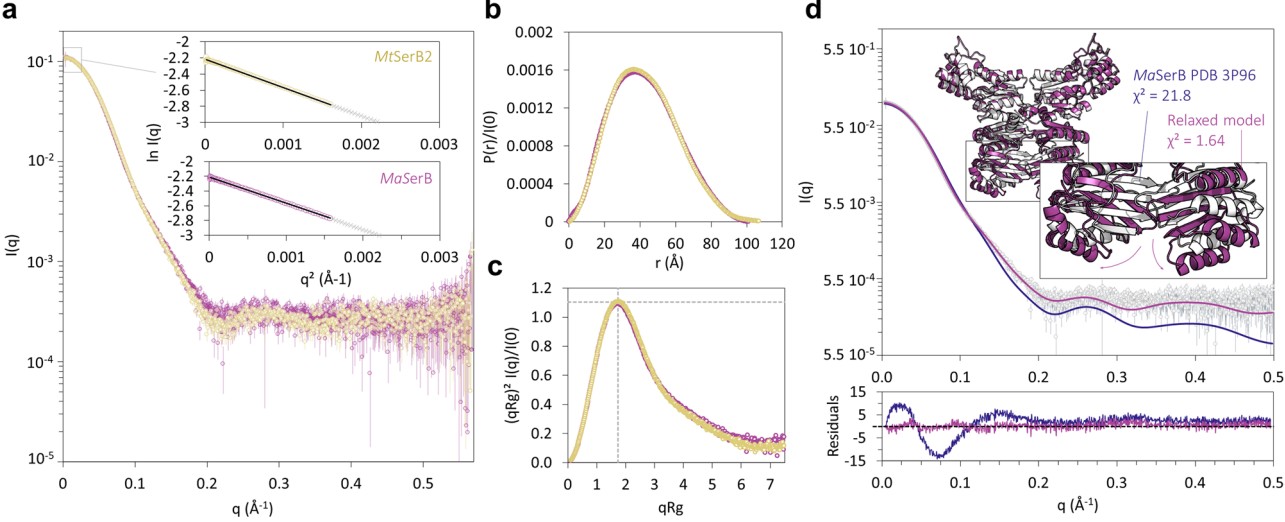

**Fig. 4 Comparison of the SAXS solution structures of *Mt*SerB2 and *Ma*SerB dimers. a** Superimposition of $I(q)$ versus $q$ as log-linear plots for *Mt*SerB2 and *Ma*SerB dimers. The inset shows the Guinier fit (coloured symbols) for $qR_g < 1.3$ with cross symbols (grey) indicating data beyond the Guinier region. Error bars represent the s.d. of averaged data (31 frames for *Mt*SerB2 and 11 frames for *Ma*SerB). **b** Dimensionless Kratky plots for the data in (**a**). **c** $P(r)$ functions from the data in (**a**) normalised to $I(0)$ for comparison purposes. **d** Fits of *Ma*SerB dimer crystal structure[14] PDB: 3P96 (dark blue) and *Ma*SerB dimer model relaxed using *DADIMODO*[76] (purple) to the experimental data. The inset shows the conformational change that occurred in the crystal structure during the relaxation procedure. The error-weighted residual difference plots for the crystal structure and the relaxed model are shown in the bottom graph. Source data are available on the FigShare repository (https://doi.org/10.6084/m9.figshare.24116571).

results and the similar SAXS signatures, we draw the conclusion that the domain-swapped butterfly-like architecture is conserved among *Mt*SerB2, *Ma*SerB, and *Mm*SerB2, and correctly describes all three homodimers in solution.

**MtSerB2's residues Gln92, Cys148, Val149, Gly150 and Ile154 are involved in tetramerization.** The multiple sequence alignment of *Mt*SerB2, *Ma*SerB and *Mm*SerB2 (Supplementary Fig. 7) shows that a notable difference in terms of physicochemical

**Table 1 Structural parameters determined by SEC-UV-SAXS for *Mm*SerB2 dimer, *Ma*SerB dimer, *Mt*SerB2 dimer, *Mt*SerB2 trimer and *Mt*SerB2 tetramer.**

|  | *Mm*SerB2 dimer | *Ma*SerB dimer | *Mt*SerB2 dimer | *Mt*SerB2 trimer | *Mt*SerB2 tetramer |
|---|---|---|---|---|---|
| Guinier analysis |  |  |  |  |  |
| $I(0)$ (cm$^{-1}$) | $0.1017 \pm 5.736\ 10^{-5}$ | $0.1096 \pm 5.991\ 10^{-5}$ | $0.0937 \pm 3.322\ 10^{-5}$ | $0.0422 \pm 7.024\ 10^{-5}$ | $0.0578 \pm 5.299\ 10^{-5}$ |
| $R_g$ (Å) | $32.32 \pm 0.03$ | $32.62 \pm 0.03$ | $32.47 \pm 0.02$ | $39.61 \pm 0.16$ | $39.89 \pm 0.06$ |
| $q_{min}$ (Å$^{-1}$) | $0.00365\ (n=1)$ | $0.00365\ (n=1)$ | $0.00365\ (n=1)$ | $0.00502\ (n=4)$ | $0.00365\ (n=1)$ |
| $qR_g$ max | 1.2972 | 1.2942 | 1.3032 | 1.0116 | 1.2919 |
| Coefficient of correlation, $R^2$ | 0.9952 | 0.9992 | 0.9989 | 0.9948 | 0.9901 |
| $P(r)$ analysis |  |  |  |  |  |
| $I(0)$ (cm$^{-1}$) | $0.1022 \pm 4.84\ 10^{-5}$ | $0.1101 \pm 5.25\ 10^{-5}$ | $0.0938 \pm 2.93\ 10^{-5}$ | $0.0422 \pm 6.21\ 10^{-5}$ | $0.06 \pm 4.34\ 10^{-5}$ |
| $R_g$ (Å) | $32.54 \pm 0.02$ | $32.84 \pm 0.02$ | $32.55 \pm 0.02$ | $39.94 \pm 0.07$ | $39.62 \pm 0.04$ |
| $D_{max}$ (Å) | 99 | 101 | 107 | 143 | 125 |
| $\chi^2$ (total estimate from GNOM) | 1.134 (0.902) | 1.245 (0.900) | 1.183 (0.831) | 0.909 (0.868) | 1.244 (0.976) |
| Molecular weight analysis |  |  |  |  |  |
| $V_p$ (nm$^3$) | 125 | 126 | 127 | 183 | 254 |
| $V_p$ MW (kDa) | 103.7 | 104.7 | 105.5 | 152.2 | 210.7 |
| Bayesian inference MW (conf. interval) (kDa) | 91.2 (84.3–95.8) | 91.2 (84.3–95.8) | 88.3 (84.3–95.8) | 130.9 (121.5–142.2) | 185.8 (162.7– 221.1) |

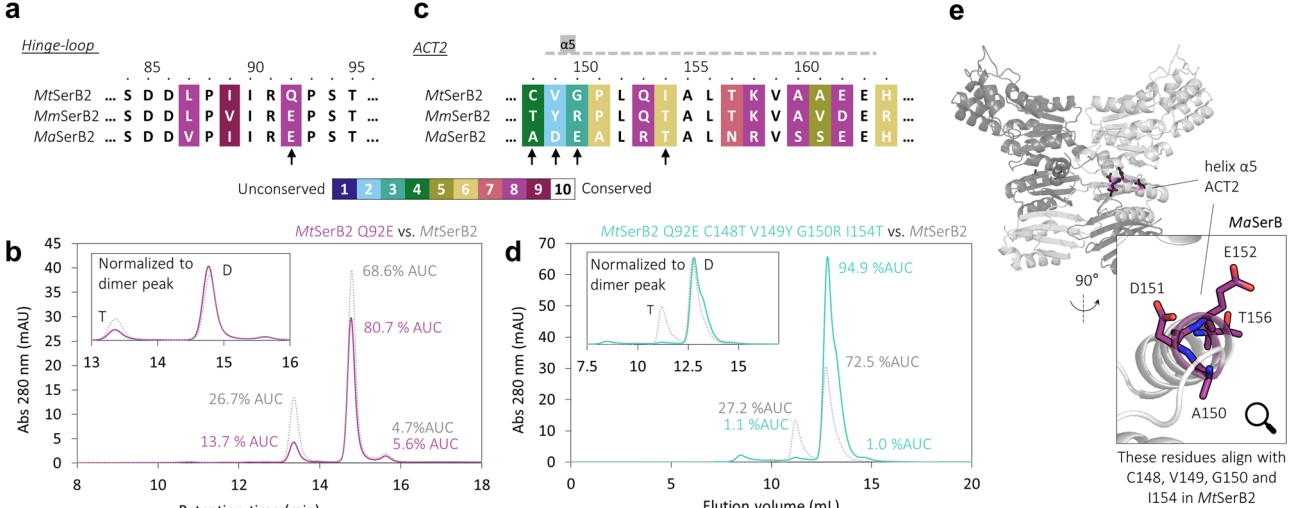

**Fig. 5 Identification of key residues in *Mt*SerB2 and analysis of their implication in tetramerization ability by mutagenesis and size-exclusion chromatography (SEC-UV). a** Sequence alignment of *Mt*SerB2, *Ma*SerB and *Mm*SerB2 focused on the hinge-loop region. The residues are coloured according to their degree of conservation between the three sequences according to the legend. The position of residue Q92 in *Mt*SerB2 is indicated by an arrow. **b** Superimposition of chromatograms obtained during the SEC-UV analysis of *Mt*SerB2 Q92E variant and native *Mt*SerB2. Area under curve percentages (%AUC) are indicated next to each peak to quantify the change in the dimer (D)/tetramer (T) ratio between the variant and the native enzyme. *Mt*SerB2 Q92E forms less tetramer than native *Mt*SerB2. A more visual indication of that change is shown in the inset where the UV absorbance at 280 nm is normalised to the dimer peak. **c** Sequence alignment of *Mt*SerB2, *Ma*SerB and *Mm*SerB2 focused on the second alpha helix (α5) of the ACT2 domain. Residues C148, V149, G150 and I154 differentiate *Mt*SerB2 from *Ma*SerB and *Mm*SerB2 by their distinct physicochemical properties (low conservation score) and their positions are indicated by arrows. **d** Superimposition of chromatograms obtained during the SEC-UV analysis of *Mt*SerB2 Q92E C148T V149Y G150R I154T variant and native *Mt*SerB2. Variant *Mt*SerB2 is essentially dimeric. **e** Three-dimensional representation of the location of residues A150, D151, E152 and T156 (homologous to key residues C148, V149, G150 and I154 for tetramerization in *Mt*SerB2) of helix α5 of the ACT2 domain in *Ma*SerB crystal structure[14] (PDB: 3P96). The residues are exposed to solvent and not engaged in intramolecular interactions. The difference in numbering comes from the fact that *Ma*SerB bears two more residues than *Mt*SerB2 at the N-term.

properties exists at the ninth residue of the hinge-loop: at pH 7.4, *Mt*SerB2 bears a neutral glutamine residue where *Ma*SerB and *Mm*SerB2 bear a negatively charged glutamate residue (Fig. 5a). As the hinge-loop region is the major determinant of domain-swapping[20,21], we wanted to probe the implication of this variation in the dimer-tetramer equilibrium that is exclusively observed in *Mt*SerB2 under the studied conditions. To this end, we constructed a *Mt*SerB2 Q92E variant by site-directed mutagenesis. Using analytical SEC-UV (Fig. 5b), we found that this variant formed less tetramer (13.7% AUC) than the native enzyme (26.7% AUC). This result indicates that the hinge-loop,

either through its flexibility or in terms of interfacial interactions, is involved in the tetramer-dimer equilibrium of *Mt*SerB2.

Mutation Q92E alone could not explain the ability of *Mt*SerB2 to tetramerize but another zone of the sequence alignment drew our attention. The N-terminal portion of the second alpha helix (α5) of the ACT2 domain is poorly conserved between the three enzymes (Fig. 5c). Comparing *Mt*SerB2 to its two exclusively dimeric orthologs, it can be noted that the nature of the residues changes significantly in this region: while *Mt*SerB2 bears rather small, neutral, hydrophobic, and apolar residues (C148, V149, G150 and I154), the residues are larger, charged, and/or more

polar in *Mm*SerB2 (T, Y, R, T) and *Ma*SerB (A, D, E, T). To test whether these residues are involved in the ability of *Mt*SerB2 to tetramerize, we further mutated *Mt*SerB2 Q92E variant by introducing the mutations C148T, V149Y, G150R and I154T. The resulting variant is more similar to *Mm*SerB2, the closest exclusively dimeric ortholog of *Mt*SerB2. A SEC-UV experiment revealed that the quintuple variant is essentially dimeric (94.9% AUC), hence confirming the involvement of the aforementioned residues in *Mt*SerB2 tetramerization (Fig. 5d). In *Ma*SerB crystallographic structure[14], the residues present at the mutated positions point to the solvent and do not seem to be engaged in intramolecular interactions stabilising the dimer (Fig. 5e). The ability of native *Mt*SerB2 to tetramerize unlike its close orthologs may thus be explained by the fact that it is energetically more favourable for the apolar and smaller residues C, V, G, and I of helix $\alpha_5$ of the ACT2 domain to be buried within protein-protein interfaces than for the solvated, larger, charged, and polar residues found at the same positions in variant *Mt*SerB2 Q92E C148T V149Y G150R I154T, *Ma*SerB and *Mm*SerB2.

**_Mt_SerB2 tetramer could be an assembly of closed monomers showing global D2 symmetry**. To get insight into the architecture of *Mt*SerB2 tetramer, we combined SEC-UV-SAXS experiments with protein-protein docking (Fig. 6). The scattering curve corresponding to the tetrameric species was acquired from a monodisperse elution peak (Supplementary Fig. 3) and differs from that of the dimer (Fig. 6a), which reflects distinct molecular shapes in solution for the two oligomeric species. The higher values obtained for the $R_g$, the $V_p$, and the $D_{max}$ determined from the *P(r)* function (Fig. 6b) indeed confirm that the tetrameric species is larger (Table 1), while the bell-shaped dimensionless Kratky plot reveals that it is also a well-folded globular protein (Fig. 6c).

Next, we designed three different types of plausible tetrameric architectures (Fig. 6d) based on the arrangement of domains within *Mt*SerB2 dimer and the fact that the ACT1 domain should retain the same relative position to ACT2 domain according to domain-swapping principles[20]: (1) dimers-of-dimers (2) tetramers of closed monomers (3) domain-swapped tetramers. The subunits to dock to create atomic tetramer models were either (1) *Mt*SerB2 dimer homology model, (2) a model of closed monomer with intramolecular ACT1-ACT2 interactions built by molecular modelling according to domain-swapping principles and equilibrated with MD simulations, or (3) one half of *Mt*SerB2 dimer homology model in which the hinge-loop is omitted and must be reconnected after the docking process. A total of 197 models were then generated using three different protein-protein docking algorithms under C2, C4 and D2 symmetry constraints. The theoretical scattering curve of each model was calculated and fitted to the experimental data using *CRYSOL*. Only six models yielded a $\chi^2$ value lower than 5.00 and were examined further. Among these, five models represented aggregate type dimers-of-dimers (Supplementary Fig. 8) that we did not consider plausible *Mt*SerB2 tetramer candidates for two reasons. First, none of them was globally symmetric with respect to the positioning of the monomeric subunits in the structure, whereas global asymmetry is a feature that has been rarely observed in oligomers to date, and usually serves specialised functions, such as 2:1 ligand binding[22–24]. Second, such dimers-of-dimers are counter-intuitive with the fact that rising the total protein concentration does not lead to a population shift towards the tetrameric species, as observed by native PAGE (Fig. 2d). The only remaining model however produced the best fit ($\chi^2$ = 2.21) to the experimental SAXS data and depicted a D2-symmetric tetramer formed by four closed monomers (Fig. 6e). This tetramer model is non-domain-

swapped, as the four monomers are closed to form an intramonomer ACT1-ACT2 interaction. The green monomer is related by C2 rotational symmetry to the yellow, pink, and grey monomers through rotation around three perpendicular C2 axes (respectively $C2_a$, $C2_b$ and $C2_c$) hence forming a complex with overall D2 dihedral symmetry. The tetrameric interface is located at the C-terminal catalytic PSP and regulatory ACT2 domains with the ACT1 domains pointing outwards from the complex. Monomers related through the $C2_a$ axis (green-yellow and pink-grey) interact via their PSP and ACT2 domains, while the monomers related through the $C2_b$ axis (green-pink and yellow-grey) interact at the ACT2 and ACT1 domains level. It is important to note that the yellow-grey (or green-pink) monomers couple is structurally distinct from the butterfly-like domain-swapped architecture adopted by *Mt*SerB2 dimer in solution: there is here no domain-swapping between monomers and there is a tilt angle between the monomers such that the monomers couple cannot be superimposed on *Mt*SerB2 domain-swapped dimer structure (Supplementary Fig. 9). A closer analysis shows that the scattering profile computed for the tetramer model correctly follows the experimental data (Fig. 6f). However, as shown by the residuals, the theoretical intensities are slightly overestimated up to q = 0.2 Å$^{-1}$. To reduce the discrepancies, we subjected the model to a SAXS-based structural relaxation. The procedure outputted a relaxed model that retained the overall D2 symmetry and architecture of the initial model (Fig. 6g). As can be seen in Fig. 6f, g, the slight repositioning of subunits relative to each other, including more pronounced displacement of ACT1 domains, were enough to significantly improve the fit at low *q* values ($\chi^2$ = 1.32). We therefore believe that the initial model, even if not accurate at the atomic scale, correctly approximates how the tetrameric form of *Mt*SerB2 is assembled in solution. Besides the fact that D2 symmetry is more observed than C4 in tetramers[22], two additional arguments give credence to the tetrameric architecture proposed here. First, residues Q92, C148, V149, G150 and I154, important to *Mt*SerB2 tetramerization as highlighted above, are located at the tetrameric interfaces (Fig. 6h). Second, in this model, the second alpha helix of the C1 cap (α7) is positioned so that it can interact with itself between two subunits (Fig. 6i). In the human ortholog *Hs*PSP, this same helix is known to unfold to allow substrates and products in and out of the active site[25] (Fig. 6j). A self-interaction as shown in the model could prevent access to the catalytic residues and explain why the tetramer is inactive.

**_Mt_SerB2 dimer specifically shifts to an extended trimer in the presence of L-Ser**. We also took a closer look at the oligomeric transition taking place in the presence of L-Ser highlighted by Yadav and Shree. We reproduced the SEC-UV experiment described in their work[9] by analysing *Mt*SerB2 in a mobile phase supplemented with L-Ser. Like the authors, we observed the appearance of a new peak, eluting before that corresponding to the dimer. The previously identified tetramer peak was still present (Fig. 7a). By repeating the experiment for increasing L-Ser to *Mt*SerB2 molar ratios, we found that the dimer peak underwent a leftward shift and deformation, while the tetramer peak only decreased slightly in area (Fig. 7b). From this observation, we concluded that the oligomeric transition originates from *Mt*SerB2 dimer. The oligomeric behaviours of *Ma*SerB and *Mm*SerB2 were also studied by SEC-UV in the presence of L-Ser under the same conditions. No dramatic peak shift was observed for either enzyme (Fig. 7c), suggesting that the oligomeric transition induced by L-Ser is specific to *Mt*SerB2. In addition, while the variants altered in their ability to tetramerize still underwent the transition, the oligomeric state of variant *Mt*SerB2ΔACT1 was left

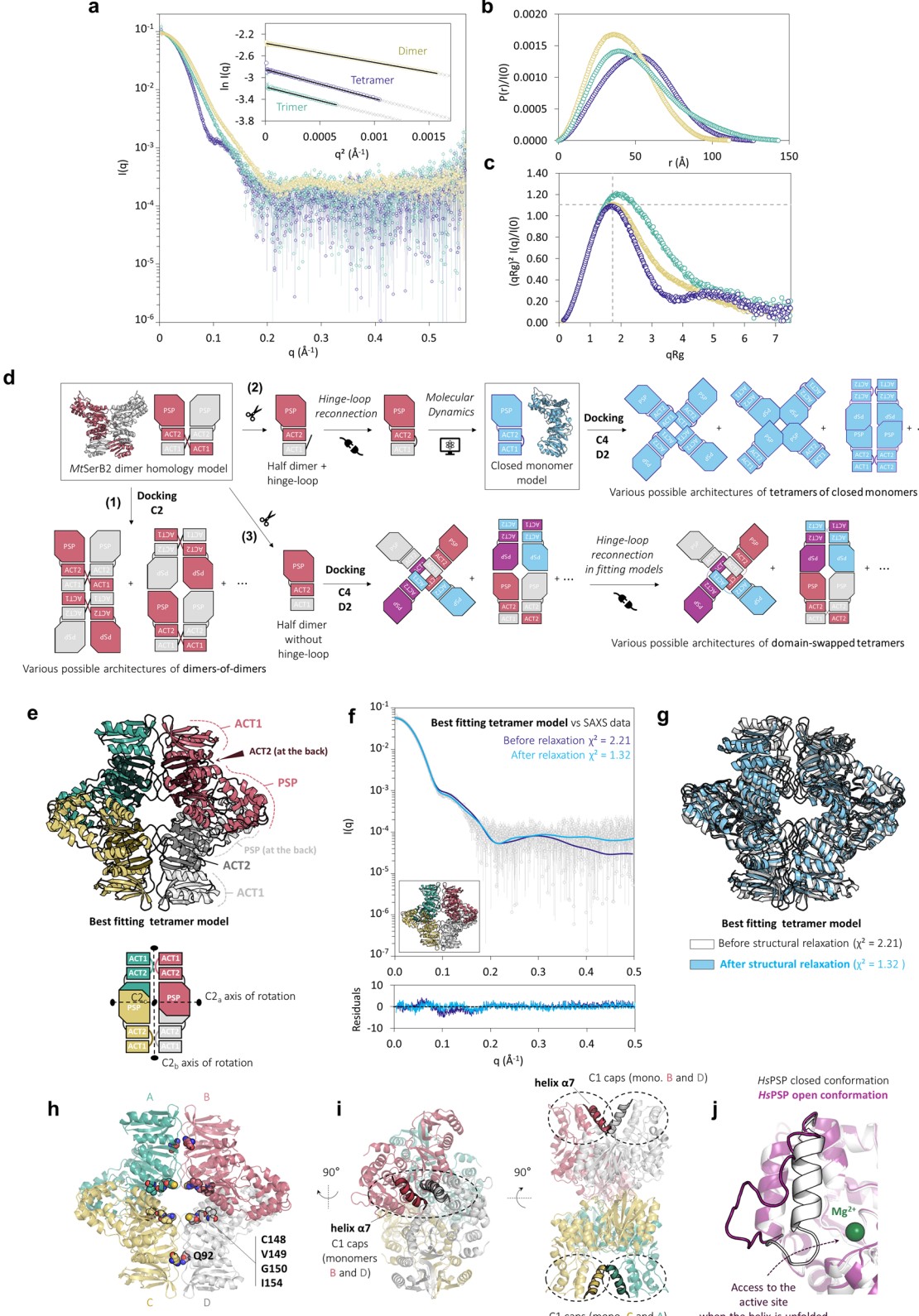

unchanged in the presence of L-Ser, which supports that the ACT1-ACT2 interface is involved in the L-Ser induced transition (Supplementary Fig. 10).

SEC-UV-SAXS was used to determine the stoichiometry of the species corresponding to the new peak appearing in the presence of L-Ser. As the latter tails off and is therefore probably composed of a mixture of species of decreasing size, special care was taken to

consider only consecutive frames for which the standard deviation on the calculated $R_g$ did not exceed 0.3 Å (Supplementary Fig. 3). This led to a SAXS curve distinct from those of the dimer and the tetramer (Fig. 6a) and from which a molecular weight value corresponding to a trimer (130.9 kDa) could be determined by the Bayesian inference approach (Table 1). The intermediate $V_p$ value also supports the trimeric stoichiometry.

**Fig. 6 Study of the architecture of *Mt*SerB2 tetramer using SEC-UV-SAXS and protein-protein docking. a** Superimposition of $I(q)$ versus q as log-linear plots for *Mt*SerB2 dimer (yellow, on absolute scale), trimer (green, offset by a factor 2.25), and tetramer (dark blue, offset by a factor 1.65). The SAXS curve have been offset by the indicated factors for visualisation and comparison purposes. The inset shows the Guinier fits (coloured symbols) for $qR_g < 1.3$ (dimer and tetramer) and $qR_g < 1.0$ (trimer) with cross symbols (grey) indicating data beyond the Guinier region. Error bars represent the s.d. of averaged data (31 frames for the dimer, 15 frames for the tetramer, and 13 frames for the trimer). **b** Dimensionless Kratky plots for the data in (**a**). **c** $P(r)$ functions from the data in (**a**) normalised to $I(0)$ for comparison purposes. **d** Strategy used for the in silico modelling of *Mt*SerB2 tetramer by symmetrical protein-protein docking. The in silico modelling steps for constructing three different types of tetrameric architectures from the space coordinates of *Mt*SerB2 dimer homology model are described schematically. **e** Cartoon representation of the best-fitting tetramer model resulting from the SAXS-based screening of models generated by protein-protein docking (top panel). Each monomer is shown in a distinct colour. The bottom panel depicts a schematic representation of the arrangement of monomers within the tetramer. The three C2 axes of rotation giving rise to the overall D2 symmetry of the tetramer are shown as dotted lines capped by an oval. C2$_a$ and C2$_b$ axes are respectively horizontal and vertical in the plane of the sheet, and C2$_c$ axis is perpendicular to the plane of the sheet. **f** Fits of the best-fitting tetramer model (**e**) resulting from the SAXS-based screening of models generated by protein-protein docking (dark blue) and of this same model relaxed using *DADIMODO*[76] (light blue) to the experimental data. The error-weighted residual difference plots for the model before and after relaxation are shown in the bottom graph. **g** Superimposition of the structures of the best-fitting tetramer model resulting from the SAXS-based screening of models generated by protein-protein docking (white) and of this same model relaxed using *DADIMODO*[76] (light blue). **h** Location of the residues involved in *Mt*SerB2 tetramerization ability (shown as spheres) at the tetrameric interfaces of the best-fitting tetramer model (**e**). **i** Two different perspectives highlighting the spatial proximity of the two dynamic helices α7 (second helix of the C1 cap) from monomers B and D, and C and A. **j** Superimposition of the crystal structures of *Hs*PSP monomers[25] in open and closed conformations (PDB: 6HYY). The focus is on residues 40–56 of the C1 cap (position of helix α7 by homology in *Mt*SerB2). The residues are folded as an alpha helix in the closed conformation (white) and the form of a flexible loop in the open conformation (pink).

Based on the SAXS data, four arguments show that *Mt*SerB2 trimer is an extended species: (1) The linear range of the Guinier plot (Fig. 6a) does not extend above a $qR_g$ value of 1.0. (2) Despite the difference in stoichiometry, the $R_g$ of the trimer (39.6 Å) is very close to that of the tetramer (39.9 Å, Table 1). (3) The $P(r)$ function (Fig. 6b) displays a longer tail and the $D_{max}$ (143 Å) is larger than that determined for the tetramer (125 Å). (4) The dimensionless Kratky plot (Fig. 6c) exhibits a maximum that exceeds 1.104 at $qR_g = \sqrt{3}$. Its bell shape is also less defined than for the dimeric and tetrameric species, which reflects a higher degree of flexibility.

**MtSerB2 dimer ACT1-ACT2 interface is likely disrupted by L-Ser.** We then sought to determine how L-Ser specifically interacts with *Mt*SerB2 dimer to trigger trimerization. To identify the residues involved, we predicted and compared ligand binding sites in *Mt*SerB2, *Ma*SerB and *Mm*SerB2 dimers in silico. The prediction highlights a pocket located at ACT1-ACT2 intersubunit interface that would exist in *Mt*SerB2 but not in its two orthologs (Fig. 7d). Interestingly, this pocket contains E33 and R103, two residues that are thought to be involved in L-Ser binding based on homology with the ACT domains of *E. coli* phosphoglycerate dehydrogenase[19].

Docking of L-Ser in the predicted specific pocket suggests that L-Ser binding is mediated by interactions involving the side chains of E33, R103 and T136 (Fig. 7e). We hypothesised that mutating these three residues to alanine would prevent the selective interaction with L-Ser and therefore disable the formation of the trimeric species. Our supposition was tested by engineering *Mt*SerB2 triple alanine variant E33A R103A T136A and by analysing its oligomeric behaviour by SEC-UV. Whether in the presence or absence of L-Ser, the associated chromatogram shows two peaks with respective retention times corresponding to the tetrameric and dimeric forms of wild-type *Mt*SerB2. In the presence of L-Ser, no leftward peak shift is observed for the variant dimer peak, in contrast to wild-type *Mt*SerB2, which shows that the oligomeric equilibrium of the triple alanine variant is insensitive to L-Ser (Fig. 7f). We therefore conclude that E33, R103 and T136 residues are the peculiar features of *Mt*SerB2 allowing the formation of the trimer in the presence of L-Ser.

Based on these results, we assumed that the interaction of L-Ser with E33, R103 and T136 could trigger the dissociation of *Mt*SerB2 dimer into monomers by disrupting the stabilising ACT1-ACT2 intermolecular interface. In turn, monomers would quickly reassociate to the observed trimeric form. Results from SAXS-based rigid body modelling of *Mt*SerB2 trimer further strengthened our hypothesis: among 10 trimer models generated under C3 symmetry constraints ($1.75 < \chi^2 < 3.52$, Supplementary Fig. 11), we found that 7 models, although not perfectly identical, shared a common architecture and were the models that best fit the data ($1.75 < \chi^2 < 2.76$). As shown in Fig. 7g, they exhibit a central core composed of the C-terminal catalytic PSP domains and ACT2 domains, while the N-terminal ACT1 domains point outwards from the structure and are separated from the ACT2 domains by the hinge-loop adopting an extended conformation. This kind of spatial arrangement is structurally consistent with the fact that L-Ser could disrupt the ACT1-ACT2 interaction and with the flexible nature of the hinge-loop linking ACT1 and ACT2 domains. A trimer mostly stabilised by the PSP-ACT2 core is also in line with the ability of variant *Mt*SerB2 ΔACT1 to form a small amount of trimer (Supplementary Fig. 6).

**MtSerB2 is inhibited by L-Ser in a partial, predominantly competitive fashion.** Finally, we assessed the effect of L-Ser as a modifier of *Mt*SerB2 dimer catalytic activity by performing steady-state kinetics experiments over a range of L-Ser concentrations. Initial dephosphorylation rates were measured at varying phosphoserine and fixed L-Ser concentrations ([L-Ser]) and plotted as shown in Fig. 8a. As initial rates decreased with increasing [L-Ser], we could conclude that L-Ser inhibited *Mt*SerB2 phosphatase activity. To identify the kinetic mechanism of inhibition, we followed the systematic approach proposed by Baici[26] (thoroughly described on the website https://www.enzyme-modifier.ch). First, apparent $k_{cat}$ and $K_M$ values could be determined by fitting the condensed form of the general modifier Eq. (2)[26] to the $v$ vs [S] data shown in Fig. 8a. Next, the inhibition mechanism was diagnosed by replotting the apparent $k_{cat}$, $K_M$, specificity constant ($k_{cat}/K_M$), and their respective multiplicative inverses versus [L-Ser] and examining the shapes of the plots (Fig. 8b). As described on Baici's website, each kinetic mechanism gives a unique combination of replots. The shape of each replot is designated by a letter, depending on how each parameter behaves when [inhibitor] increases (Supplementary Table 1): independent (A, D, H, L, O); increases hyperbolically (B, E, I, M, P); decreases hyperbolically (C, F, J, N, Q) or increase

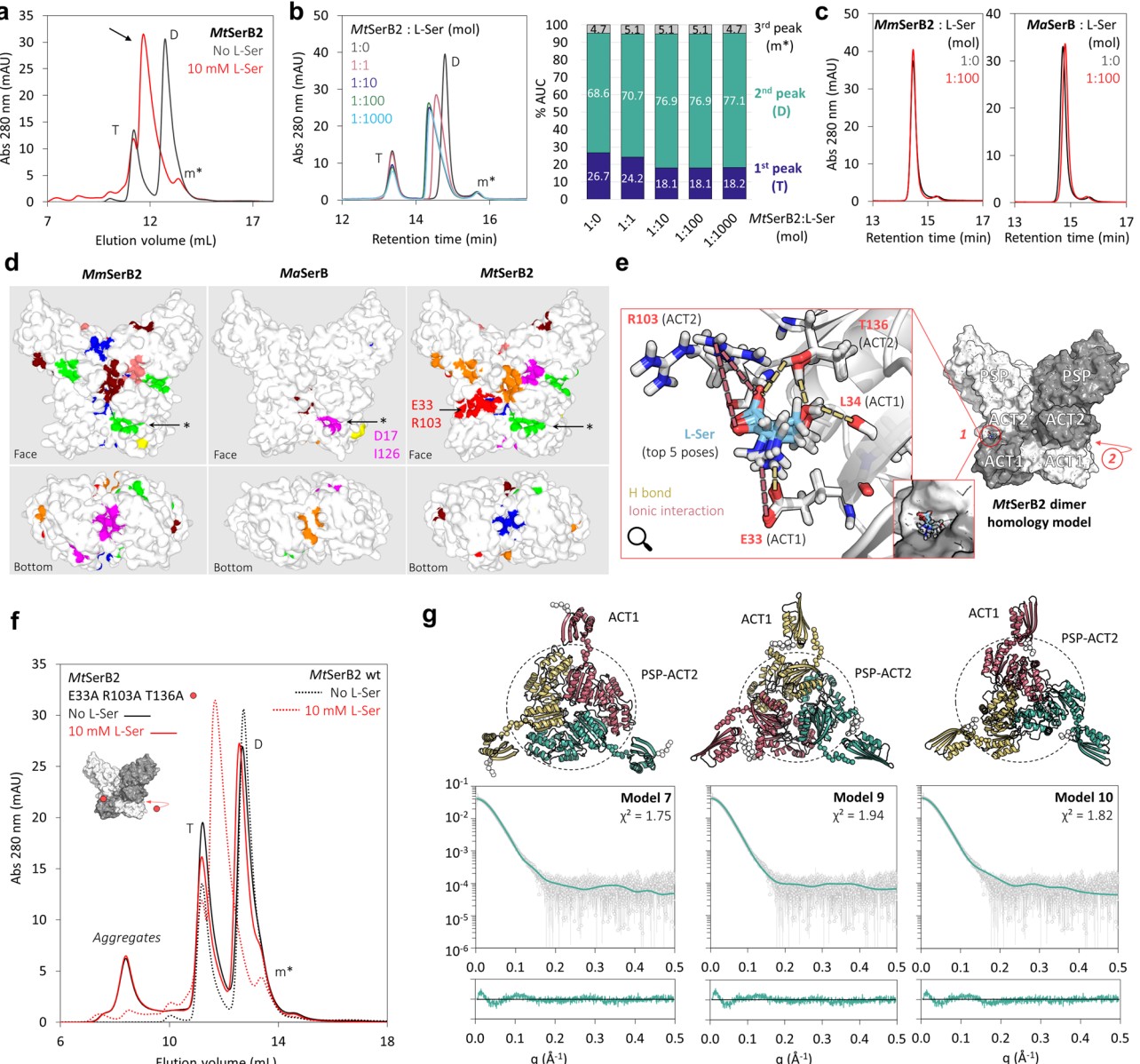

**Fig. 7 Study of the dimer-to-trimer transition induced by L-Ser in *Mt*SerB2 through size-exclusion chromatography (SEC-UV), docking, mutagenesis and SAXS-guided modelling. a** Superimposition of the chromatograms obtained during the SEC-UV analysis of *Mt*SerB2 in the absence (black) and presence of 10 mM L-Ser (red) in the mobile phase. The arrow highlights the appearance of a new peak in the presence of L-Ser. T tetramer, D dimer, m* probable monomer. **b** The left panel shows the superimposition of the chromatograms obtained during the SEC-UV analysis of *Mt*SerB2 in the presence increasing L-Ser amounts in the mobile phase. The histogram in the right panel shows the change in percentage area under the curve for each peak of the chromatograms in the left panel as a function of *Mt*SerB2 to L-Ser molar ratio. **c** Superimposition of the chromatograms obtained during the SEC-UV analysis of *Mm*SerB2 (left panel) and *Ma*SerB (right panel) for enzyme to L-Ser molar ratios of 1:0 (black) and 1:100 (red) in the mobile phase. **d** Ligand binding sites in *Mm*SerB2, *Ma*SerB and *Mt*SerB2 predicted by *P2Rank*. The arrows and asterisks indicate L-Ser binding pockets in the three orthologs as expected on the basis of *Ma*SerB-L-Ser cocrystal structures where L-Ser interacts with residues D17 and I126 (PDB: 5JLR[27] and 5JLP[28]). The red pocket indicated by an arrow and the label E33 R103 is only predicted in *Mt*SerB2. **e** Three-dimensional representation of the top five poses in terms of docking score for the induced-fit docking of L-Ser at pH 7.4 in the ACT1-ACT2 interfacial binding pocket of *Mt*SerB2 dimer homology model containing residues E33 and R103 (see **d**). The arrow labelled with the number 2 indicates the same pocket, located on the other side of the enzyme by C2 symmetry. **f** Superimposition of the chromatograms obtained during the SEC-UV analysis of *Mt*SerB2 triple alanine variant E33A R103A T136A in the absence (plain black trace) and presence of 10 mM L-Ser (plain red trace) in the mobile phase, and of wild-type *Mt*SerB2 in the absence (dotted black trace) and presence of 10 mM L-Ser (dotted red trace) in the mobile phase. The triple alanine variant dimer (D) does not undergo any oligomeric transition (leftward peak shift) in the presence of L-Ser. **g** Cartoon representation of the three best-fitting *Mt*SerB2 trimer models obtained by SAXS-based rigid body modelling. The models have a common architecture where the ACT1 domain points to the outside of the assembly while the PSP and ACT2 domains compose the core. The fit of each model to the experimental data as calculated by *CRYSOL* as well as the corresponding error-weighted residual difference plots are shown in the bottom panel. Error bars represent the s.d. of averaged data (13 frames).

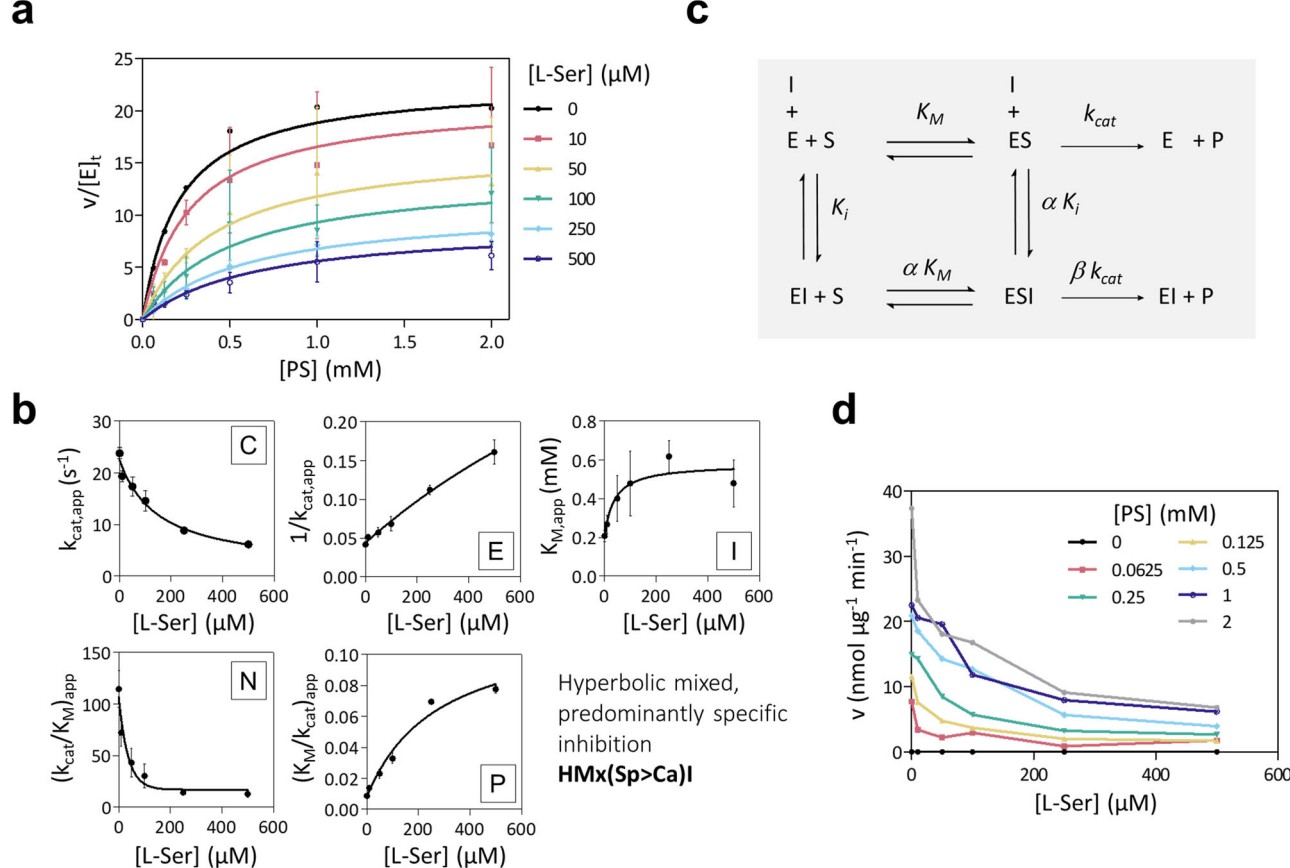

**Fig. 8 Evaluation of the effect of L-Ser as a modifier of the kinetics of phosphoserine (PS) dephosphorylation by *Mt*SerB2 dimer. a** Plots of initial velocities (nmol phosphate released per μg of *Mt*SerB2 per minute) as $v/[E]_t$ (min$^{-1}$) versus substrate concentration for PS dephosphorylation by *Mt*SerB2 dimer fraction pool at various fixed L-Ser concentrations (μM). Error bars represent the s.d. of three experiments. Plain lines are the fit of the developed form of the general modifier Eq. (2) to the experimental data. **b** Dependence of the apparent kinetic parameters $k_{cat}$, $K_M$, $(k_{cat}/K_M)$ on L-Ser concentration. The shape of the plots are designated by a framed letter (as explained in Supplementary Table 1 and https://www.enzyme-modifier.ch) and allows the identification of a mechanism of hyperbolic mixed, predominantly specific inhibition according to the methodology of Baici[26]. Plain lines are the fits of the dependency of apparent kinetic parameters on [L-Ser] to the experimental data based on the general modifier equation (Eqs. (3–7)). Error bars represent the standard error of the parameter as calculated by Prism following non-linear regression. **c** General modifier mechanism. E enzyme, S substrate (phosphoserine), P product (phosphate), I inhibitor (L-Ser), $k_{cat}$ catalytic constant, $K_M$ Michaelis constant, $K_i$ inhibition constant, α reciprocal allosteric coupling constant, β factor by which the modifier affects $k_{cat}$. **d** Plot of initial velocity versus L-Ser concentration at fixed PS concentrations. Source data are available on the FigShare repository (https://doi.org/10.6084/m9.figshare.24116571).

linearly (G, K, R). For *Mt*SerB2, the shape of the parameters dependencies on [L-Ser] were found to match the combination C-E-I–N-P (Fig. 8b). This combination corresponds to a hyperbolic mixed, predominantly specific inhibition ('HMx(Sp>Ca)I' according to Baici nomenclature) mechanism, which, in more common terms, refers to a mechanism of partial, predominantly competitive inhibition. In this mechanism (Fig. 8c), L-Ser can bind to the enzyme-substrate complex but with a lower affinity (α > 1) than to the free enzyme (predominantly competitive character). The ternary complex formed between the enzyme, the substrate and L-Ser is still able to form a product, but at a slower rate (β < 1, partial character). This behaviour agrees with Grant's conclusions[19]. The partial nature of the inhibition was slightly ambiguous in view of the quasi-linear dependence of apparent $1/k_{cat}$ on [L-Ser] but was confirmed by the plot of initial velocities versus L-Ser concentration showing a plateau rather than reaching zero (Fig. 8d). Fitting the developed form of the general modifier Eq. (2) to the data yielded a $K_i$ value of 22.0 ± 5.3 μM matching the reported value (19 ± 2 μM[19]), a value of 3.52 ± 1.20 for parameter α translating a 2–5 times lower affinity of phosphoserine for L-Ser-*Mt*SerB2 complex, and a value of 0.0006 ± 0.0004 for parameter β reflecting an almost

zero turnover rate for L-Ser-*Mt*SerB2-phosphoserine complex ($\beta k_{cat} = 0.01$ s$^{-1}$). These results are in line with the formation of a weakly active trimer upon disruption of the dimeric assembly. L-Ser would interact more easily with the free dimer but could also disrupt *Mt*SerB2-phosphoserine complex. Our trimer models, whose interfaces formed at the PSP domains could restrict but not totally prevent access to the active site, explain the remaining low activity.

In addition, using the same methodology, we diagnosed L-Ser as a total parabolic mixed, predominantly uncompetitive (or 'S-linear I-parabolic noncompetitive') inhibitor of *Ma*SerB and *Mm*SerB2 (Supplementary Fig. 12). In this mechanism, L-Ser can bind both the free enzyme and the enzyme-substrate complex (with more affinity), and a second L-Ser molecule then binds the enzyme-substrate-L-Ser complex to totally inhibit enzyme activity. On the basis of *Ma*SerB-L-Ser cocrystal structures (PDB: 5JLR[27] and 5JLP[28]), we hypothesize that L-Ser would interact with residues D17 and I126 at the ACT1-ACT2 domain interface (Fig. 7d) and that a second binding site leading to total inhibition would appear after a conformational change triggered by substrate binding. All in all, this distinct inhibitory mechanism, involving the binding of two L-Ser molecules in the close

orthologs enzyme-substrate complexes, further testifies to the peculiarity of the interaction of L-Ser with *Mt*SerB2.

## Discussion

Inhibiting the phosphatase activity of the PSP *Mt*SerB2 from *M. tuberculosis* is a promising approach for the development of new antituberculosis agents. However, designing compounds targeting the active site might not be an optimal strategy in terms of selectivity, since *Mt*SerB2 and the human ortholog *Hs*PSP share highly conserved catalytic pockets[29]. Nevertheless, the original and emerging approach that represents the targeting of protein self-association can be applied to *Mt*SerB2 to circumvent the limitation[30]. Indeed, *Hs*PSP forms an elongated homodimer[31,32] while prior studies have described *Mt*SerB2 in solution as a distinct, active domain-swapped homodimer that undergoes a quaternary structure change to an inactive higher-order homomeric species, believed to be a tetramer, in the presence of its endogenous allosteric feedback inhibitor L-Ser[9,19]. Based on this postulate, we aimed to deepen the structural knowledge regarding *Mt*SerB2 to offer a rational basis for the design of allosteric inhibitors targeting quaternary structures.

Our investigations lead us to propose the oligomeric equilibrium model depicted in Fig. 9. The results first revealed that in the absence of L-Ser, *Mt*SerB2 co-exist as two different main oligomeric forms in solution: (1) a majority ACT1 domain-swapped butterfly-like homodimer (Fig. 9a), and (2) a tetramer that may be formed by the interaction between four closed monomers under D2 symmetry (Fig. 9b). While we confirmed the architecture of the dimer through direct comparison with the SAXS solution structure of a close ortholog of known crystallographic structure (*Ma*SerB[14], 83.7% sequence identity) and interface analysis, we gained insight into the architecture of the tetramer through SAXS-guided symmetric protein-protein docking. The closed monomer model used in the docking experiment that led to the most convincing tetramer model was built according to the fundamentals of domain-swapping so that the ACT1 domain is relocated below the ACT2 domain to form

an eight-stranded beta-sheet ACT dimer (Fig. 9c). Given the spatial arrangement of the two oligomeric species, and the presence of a rapidly oligomerising monomer in solution suspected by SEC-UV-MALS, we believe that *Mt*SerB2 tetramer-to-dimer conversion requires the dissociation of the tetramer into closed monomers, themselves in rapid equilibrium with an open form (Fig. 9d) able to dimerise through the retro-exchange of the ACT1 domain. While this process occurs on a scale of hours to days, the dimer-to-tetramer conversion was observed to take months. From a thermodynamics and kinetics point of view, it is probably easier to break the interfacial interactions between four closed monomers, whose conformation gives them a certain stability in solution, and to overcome the kinetic barrier for their opening than to disrupt a highly stable intertwined dimer into open monomers of high energy (Fig. 9e). This mechanism is in line with what has been proposed by Eisenberg and its co-workers for the monomer–dimer interconversion of proteins swapping entire domains[33]. Assuming that *Mt*SerB2 folds into a closed monomer after ribosomal translation, such an energy landscape, combined to definite sequence property shifts (residues C148, V149, G150 and I154) compared to exclusively dimeric orthologs *Ma*SerB and *Mm*SerB2, explains why the enzyme also forms a minority amount of tetramer besides a stable intertwined dimer.

Biophysics, mutagenesis, docking, and kinetics also helped us decipher the peculiar regulatory interaction between *Mt*SerB2 dimer and L-Ser. While *Ma*SerB and *Mm*SerB2 are likely feedback inhibited at the active site level through the propagation of conformational changes following the allosteric binding of two L-Ser molecules to the enzyme-substrate complex, *Mt*SerB2 shows a distinct mechanism. Our findings suggest that L-Ser disrupts the ACT1-ACT2 dimer interface by interacting with residues E33, R103 and T136, which causes the dissociation of the dimer into monomers unable to re-establish intra- or intermolecular ACT1-ACT2 interactions (Fig. 9f). Not stable as such, the monomers would associate in the form of a trimer of very low catalytic activity explained by hindered access to the active site due to the trimer PSP-ACT2 core interface (Fig. 9g).

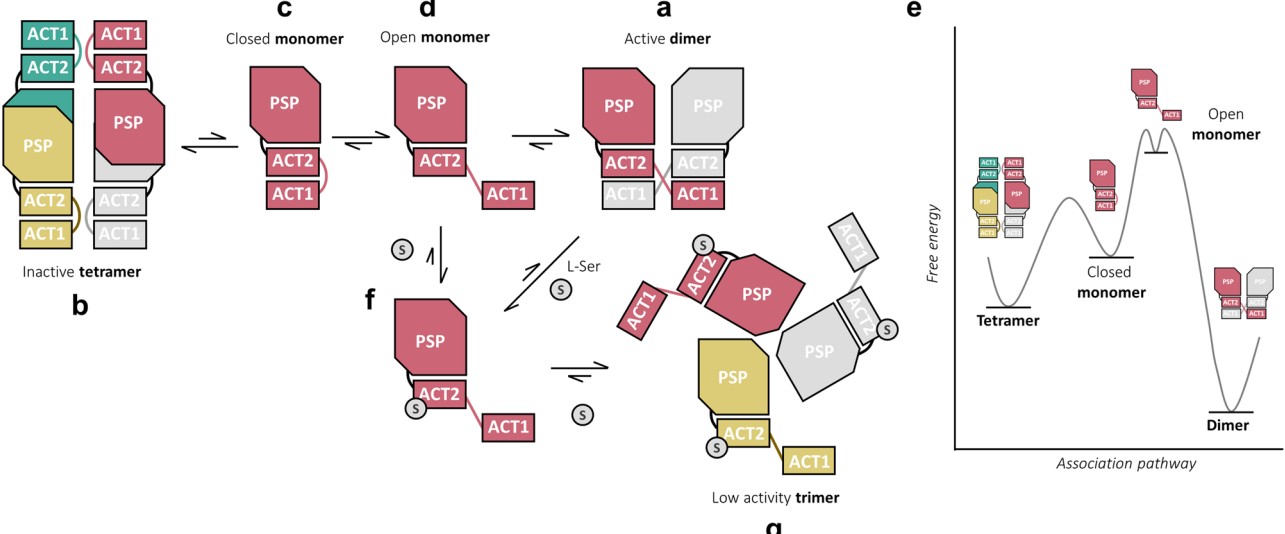

**Fig. 9 *Mt*SerB2 quaternary structure equilibrium model.** *Mt*SerB2 active ACT1 domain-swapped dimer (**a**) is in equilibrium with an inactive tetramer formed of four closed monomers in which ACT1 and ACT2 domains interact intramolecularly (**b**). The dimer-tetramer interconversion proceeds through dissociation of the oligomers into conformationally flexible monomers that can adopt a closed pro-tetramer conformation (**c**) and an open pro-dimer conformation (**d**). The relative free energy of each species is estimated in the qualitative free energy diagram (**e**) shown in the right panel. L-Ser disrupts intermolecular ACT1-ACT2 domains interfaces in *Mt*SerB2 dimer, leading to dissociation into monomers stabilised in their open conformation by the interaction with L-Ser (**f**). These monomers rapidly combine to form a trimer of very low activity (**g**) stabilised through intermolecular interactions at PSP-ACT2 domains.

We showed that *Mt*SerB2 exists as alternative quaternary forms of distinct stoichiometries and activity levels interconverting through dissociation into a conformationally flexible subunit that can adopt alternative tertiary structures dictating the higher-order oligomer to be formed. The distinct activity levels of the oligomers could be structurally interpreted by facilitated access to active site in the dimer and hindered access in the trimer and tetramer. These peculiar properties are those that define the homo-oligomeric proteins called "morpheeins"[34]. The morpheein concept was first described in 2005 by Tang et al. with the example of the hexameric and octameric forms of porphobilinogen synthase[35]. In addition to shifting the classical one-sequence-one-structure-one-function paradigm, morpheeins also provide a novel model for allosteric regulation. The activity of a morpheein can indeed be controlled by the displacement of the equilibrium towards oligomeric forms of distinct activity levels in response to physiological stimuli, such as increase in ligand concentration. The control of catalysis is structurally explained by a quaternary structure change that modulates active site access[36,37].

However, the morpheein model of allostery is originally defined by a mechanism of conformational selection in which a stabilising ligand binds to one assembly, not the others, and displace the equilibrium towards this assembly. It does not imply a ligand-induced disruption of one assembly. Our results therefore suggest that the definition of morpheeins could potentially be extended to include allostery by oligomeric disruption and hence encompass a larger repertoire of enzymes.

Consequently, such regulatory properties can serve as a basis for allosteric drug discovery[38–40]. New potential targets arise from the morpheein behaviour we propose for *Mt*SerB2, as the enzyme PSP activity could be inhibited by compounds designed to disrupt the active dimer or stabilise the (nearly) inactive trimeric and tetrameric forms.

Moreover, the morpheein character of *Mt*SerB2 is also consistent with the work of Shree et al., who reported that *Mt*SerB2 appeared to possess, in addition to its metabolic function within *Mtb*, an effector function in host cells by interacting with cytoskeleton components, anti-apoptotic proteins, and modulating the expression of various genes[10]. Proteins like *Mt*SerB2 in which one polypeptide chain performs more than one physiologically relevant function are referred to as 'moonlighting'[41–43]. Some moonlighting proteins are in fact morpheeins, with alternative functions arising from different oligomeric states[44,45].

Finally, our results highlighted that near-identical orthologs of *Mt*SerB2 (>83% sequence identity) exhibited a different oligomeric behaviour and allosteric response to endogenous ligand binding. This constitutes another example that high percentages of sequence identity can still lead to distinct oligomeric behaviours[46–49], which emphasize the importance of experimentally assessing stoichiometries, architectures and dynamics along with homology modelling and ab initio structure prediction methods. The particular properties of *Mt*SerB2 are proof that the usual confidence in a homology-based structure model beyond 30% sequence identity[50] is not sufficient to draw an exhaustive portrait of the target protein quaternary structures. Moreover, mapping the evolutionary trajectory of mycobacterial PSPs could bring further insights regarding the spontaneous evolution of allosteric regulation mechanisms. Adding *Mt*SerB2 relative to its non-morpheein orthologs could indicate whether the morpheein mode of allostery is an ancestral strategy for the regulation of L-Ser biosynthesis or if, conversely, it is a more recent and sophisticated one.

We believe that further structural investigation on *Mt*SerB2 will create exciting avenues for a structure-based drug design approach targeting the oligomeric interfaces of *Mt*SerB2, as well as for a deeper understanding of the multifunctional properties of *Mt*SerB2 and the metabolic control of L-Ser biosynthesis in *Mtb*.

## Methods

**Plasmid material and mutagenesis**. All plasmids used in this work are based on the AVA0421 vector described by Choi et al.[51]. The plasmids respectively encoding *Mt*SerB2, *Ma*SerB and *Mm*SerB2 were directly provided by the SSGCID. To construct the plasmids encoding *Mt*SerB2 ΔACT1 and *Mt*SerB2 Q92E, the pAVA0421-*Mt*SerB2 was amplified using the primers listed in Supplementary Table 2 according to the one-step site-directed mutagenesis strategy of Liu et al.[52]. The plasmids encoding *Mt*SerB2 Q92E C148T V149Y G150R I154T and *Mt*SerB2 E33A R103A T136A were manufactured by GenScript with special attention to exactly match the pAVA0421 construction for reproducibility purposes. Information on the complete expressed sequences of the studied enzymes can be found in Supplementary Table 3.

**Expression and purification of recombinant proteins**. The plasmids were transformed into *E. coli* BL21 (DE3) pLysS thermocompetent cells. The transformed cells were grown at 37 °C in LB medium containing 100 mg/mL ampicillin and 34 mg/mL chloramphenicol. Protein expression was induced at $OD_{600}$ 0.6–0.8 by the addition of 0.2 mM isopropyl β-D-1-thiogalactopyranoside (IPTG) and the cells were further grown for 18 h at 20 °C. Cells were harvested by centrifugation ($2500 \times g$, 4 °C, 30 min), resuspended in lysis buffer (50 mM Tris-HCl pH 7.4, 300 mM NaCl, 1 mg/mL lysozyme, cOmplete EDTA-free protease-inhibitor cocktail from *Roche*), and disrupted by sonication over ice (six cycles of 30 s at 20 W with 30 s of rest on ice in between). The soluble protein fraction was recovered by centrifugation ($24,000 \times g$, 4 °C, 1 h) and loaded onto a 5-mL HisTrap FF crude column filled with nickel (*Cytiva*) connected to an AKTA Purifier 10 FPLC system. Unbound protein was washed away with buffer A (50 mM Tris-HCl pH 7.4, 300 mM NaCl, 30 mM imidazole). Bound protein was then eluted in buffer B (50 mM Tris-HCl pH 7.4, 300 mM NaCl, 200 mM imidazole). Fractions containing the eluted protein were pooled and exchanged for buffer A2 (50 mM Tris-HCl pH 7.4, 150 mM NaCl, 1 mM tris(2-carboxyethyl)phosphine (TCEP)). To remove the hexahistidine tag, His6-HRV 3C protease was added to the protein solution (1 µg per 200 µg protein) and the mix was incubated overnight at 4 °C. To separate the cleaved protein from the free hexahistidine tag and protease, the protein solution was reapplied to a 1 mL HisTrap FF crude column (*Cytiva*) in buffer A2. The cleaved protein was recovered in the flow through and concentrated up to about 1 mg/mL or to higher concentrations depending on the downstream use. The concentration was determined by a measure of absorbance at 280 nm using the extinction coefficient reported in Supplementary Table 3. The purity of the protein was assessed by SDS-PAGE using a 12% polyacrylamide gel and Coomassie Brilliant Blue R staining. The protein solution was finally aliquoted and flash-frozen in liquid nitrogen for storage at −80 °C.

**Native PAGE**. Native PAGE analyses were conducted on 10% polyacrylamide 0.4 M Tris-HCl pH 8.8 gels. The samples to load consisted of a few micrograms of protein (usually 5 µg) mixed with 5 µL 4× sample buffer (0.12 M Tris-HCl pH 6.8, 0.008% bromophenol blue, 30% glycerol). Migration was performed at 110 V and room temperature for 80 min in a 0.2 M glycine 0.025 M Tris buffer. The gels were stained with Coomassie Brilliant Blue R.

**Mass photometry**. The landing of the protein species present in *Mt*SerB2 samples was recorded using a Refeyn Two MP instrument (*Refeyn Ltd*) by adding 1 μL of a diluted protein stock solution (230 nM) directly into a 19 μL drop of filtered buffer (50 mM Tris-HCl pH 7.4, 150 mM NaCl). Movie acquisition was performed during 60 s with the DiscoverMP software (version 2022 R1, *Refeyn Ltd*) and data were analysed using the default settings. Prior to the experiments was performed a contrast-to-mass calibration using a solution of protein standards with molecular weights of 66, 146, 480 and 1048 kDa.

**Semi-preparative SEC**. Protein samples were analysed and/or separated using either a Superdex 200 10/300 GL column (*GE Healthcare*) for *Mt*SerB2 (ΔACT1), *Mm*SerB2, and *Ma*SerB, or a Superdex 200 Increase 10/300 GL column (*Cytiva*) for *Mt*SerB2 Q92E C148T V149Y G150R I154T and *Mt*SerB2 E33A R103A T136A. The columns were connected to an AKTA Purifier 10 FPLC system. Data were recorded and processed using Unicorn 5.11 software (GE Healthcare). Samples (100–250 μL) containing 0.75-3.75 mg protein (pre-incubated with the desired L-Ser concentration where applicable) were injected onto the column and analysed in buffer A2 (50 mM Tris-HCl pH 7.4, 150 mM NaCl, 1 mM TCEP) or buffer A2Ser (50 mM Tris-HCl pH 7.4, 150 mM NaCl, 1 mM TCEP, 10 mM L-Ser) at a flow rate of 0.2 mL min$^{-1}$ (Superdex 200 10/300 GL) or 0.5 mL min$^{-1}$ (Superdex 200 Increase 10/300 GL). The eluate was fractionated by 250 μL when the sample was to be recovered. Fractions homogeneous in terms of oligomerisation state were pooled together after native PAGE analysis.

**SEC-UV-MALS**. Protein samples were analysed using a BioResolveSEC mAb 200 Å 2.5 μm 7.8 × 300 mm column (*Waters*) preceded by a BioResolveSEC mAb 200 Å 2.5 μm 4.6 ×30 mm precolumn (*Waters*) mounted on a LC 1260 Infinity II Bio-Inert (*Agilent*) HPLC system connected in-line to a 1260 Infinity II Bio-SEC Multi-Detector system equipped with a MDS LS dual angle (15°/90°) light scattering detector and a MDS DLS dynamic light scattering detector (90°). The detectors were normalised to 5 mg mL$^{-1}$ BSA as recommended per the manufacturer (MW = 66,463 g mol$^{-1}$, ε = 0.670 mL mg$^{-1}$ cm$^{-1}$, dn/dc = 0.186 mL g$^{-1}$). Data were recorded and processed using Bio-SEC software (*Agilent*). Samples (10–30 μL) containing 10–50 μg protein (pre-incubated with the desired L-Ser concentration where applicable) were injected onto the column and analysed at a flow rate of 0.5 mL min−1 in SEC buffer (50 mM Tris-HCl pH 7.4, 150 mM NaCl) or SEC buffer containing L-Ser. The desired concentration of L-Ser in the mobile phase was obtained through the mixing system of the HPLC by combining SEC buffers with L-Ser (50 mM Tris-HCl pH 7.4, 150 mM NaCl, 1 mM TCEP, 2.5 or 25 mM L-Ser) and without L-Ser. The weight averaged molar mass was determined for each protein species eluting as a monodisperse peak at 280 nm, using a dn/dc ratio of 0.183 mL g$^{-1}$ for *Mt*SerB2 or a default value of 0.185 mL g$^{-1}$ for the other enzymes.

**SEC-UV-SAXS data collection and treatment**. SEC-UV-SAXS experiments were carried out on the SWING beamline at SOLEIL Synchrotron (Saint-Aubin, France). The X-ray wavelength (λ) was set to 1.033 Å and the sample to the detector (162.5 × 155.2 mm² EigerX4M detector) distance was set to 2000 nm. Those parameters corresponded to a scattering wave-vector range of 0.0036 Å$^{-1}$ < q < 0.5 Å$^{-1}$, where q = 4π sin θ/λ and 2θ is the scattering angle. The sample solutions were circulated in a thermostated quartz capillary with a diameter of 1.5 mm and a wall thickness of 10 μm inserted in a vacuum

chamber. All protein samples were thawed at room temperature and centrifuged for 5 min at 6000×*g* before the SEC-UV-SAXS experiment. For each analysis, samples (50 μL) containing 0.46–0.73 mg protein (pre-incubated with the desired L-Ser concentration where applicable) were injected onto a BioResolveSEC mAb 200 Å 2.5 μm 7.8 × 300 mm column (*Waters*) preceded by a BioResolveSEC mAb 200 Å 2.5 μm 4.6 × 30 mm 229 precolumn (*Waters*) pre-equilibrated with buffer A2 or A2Ser (50 mM Tris-HCl pH 7.4, 150 mM NaCl, 1 mM TCEP, 0 or 10 mM L-Ser). The column was mounted on an Agilent HPLC system allowing for the elution of the samples at a controlled flow rate of 0.5 mL min$^{-1}$ and a temperature of 10 °C. The elution was monitored at 280 nm by a UV-diode array detector installed just downstream of the column, before the SAXS flow cell where the sample was exposed to X-rays. A total of 1140 scattering patterns were collected during the elution of the samples, with a frame duration of 1 s. The scattering signal of the buffer was collected in 180 frames before the void volume. To generate individual 1D curves, the frames were radially averaged, divided by the transmitted intensity and normalised to absolute units with water scattering as a reference using the image analysis software Foxtrot (courtesy of SWING beamline). The software was also used to generate plots corresponding to *I(0)* and *R$_g$* as a function of frames. Curves from consecutive images corresponding to the analysis of a single protein species and showing similar *R$_g$* (±0.2 or 0.3 Å) were averaged and the same operation was performed for the buffer. The averaged buffer scattering curve was subtracted from the averaged sample scattering curve to generate the final SAXS curve to be analysed. The final SAXS curves were processed and analysed using BioXTAS RAW 2.1.1 software[53]. Guinier analysis was first performed for each final SAXS curve to detect signs of interparticle interaction. In the absence of such signs, *I(0)* and *R$_g$* parameters were calculated from the Guinier plot and the curve was further analysed by determining MW from Porod volume[54] and the Bayesian inference method[55], performing Kratky analysis and calculating the *P(r)* function as recommended in BioXTAS RAW documentation.

**Sequence alignment and homology modelling**. Multiple sequence alignment was performed using PRALINE[56] online tool (https://www.ibi.vu.nl/programs/pralinewww/). *Mt*SerB2 and *Mm*SerB2 dimers were modelled using SWISS-MODEL[57]. On the only basis of the primary sequences, the algorithm proposed a homodimer model based on *Ma*SerB crystallographic structure 3P96[14] for *Mt*SerB2 and a homodimer model based on *Ma*SerB 5JJB structure[58] for *Mm*SerB2. These models were used for the representation of *Mt*SerB2 and *Mm*SerB2 in this work and for the subsequent modelling steps.

**In silico modelling of *Mt*SerB2 monomer and Molecular Dynamics (MD)**. *Mt*SerB2 theoretical closed monomer was modelled starting from *Mt*SerB2 dimer homology model. The corresponding PDB file was modified using PyMOL (2006 DeLano Scientific LLC) to keep only the coordinates of the ACT2-PSP part (H96- D398) of chain A and the ACT1-hinge part (A3-T95) of chain B. The hinge-loop of chain B was then reconnected to chain A using the 3D builder function of Maestro 11.9.011 software (Schrödinger) and minimisation through MacroModel with OPLS3e as the force field and constraining the distance between T95 and H96. This manipulation allowed the generation of a new PDB file with all residues belonging to the same chain that was used as the starting point for MD simulation. The MD simulation was run using GROMACS 2020[59] with CHARMM27 force field[60] and CMAP corrections for the protein. The protocol was based on that of Mirgaux et al.[61]. Hydrogen

atoms were added using GROMACS, and solvation was accounted for using all-atom TIP3P and coarse-grained SIRAH water particles[62–64]. A cubic box was built around the protein with at least 2.0 nm between the box edges and the protein atoms. TIP3P water molecules were placed in a 1.0-nm-thick shell around the molecular system. Coarse-grained SIRAH water particles were then placed between this shell and the edges of the box. Sodium ions were randomly placed in the bulk of the SIRAH water particles to neutralise the total charge of the system. The optimisation and MD trajectories were generated under the particle mesh Ewald periodic boundary conditions. A cut-off value of 1.2 nm was applied for Coulomb and van der Waals interactions. Temperature and pressure were respectively fixed using the Parrinello–Rahman[65] and V-Rescale algorithms[66]. Covalent bonds involving H atoms were constrained using the LINCS algorithm[67]. The resulting system was optimised using the steepest-descent algorithm for a maximal number of 2500 steps with an initial step size of 0.05 nm. During the equilibration stage of the system, the temperature was progressively increased from 50 to 310 K using short MD runs. The first run consisted of a 10 ps simulation at 50 K on the system obtained after optimisation. Afterwards, the system was relaxed for two runs of 20 ps at 150 and 310 K. Finally, a run of 50 ps at 310 K and 1 bar was performed to finalise the relaxation of the system. The equilibration was extended for 60 ns with a time step of 2 fs at 310 K and 1 bar. The production step was run for 200 ns ($100 \times 10^6$ steps) with a time step of 2 fs. The evolution of the system during the equilibration and production stages was followed through a r.m.s.d. profile (see Data Availability). The structure after 200 ns of simulation was extracted and used for the next modelling steps.

**In silico modelling of *Mt*SerB2 tetramer using symmetrical protein-protein docking**. *Mt*SerB2 tetramer models were generated by protein-protein docking under symmetry constraints using M-ZDOCK[68], ClusPro[69–72] and GalaxyTongDock[73]. Depending on the type of architecture to be modelled, a symmetry of 2 or 4 was selected on M-ZDOCK, GalaxyTongDock-C was used with C2 and C4 symmetries or GalaxyTongDock-D was used with D2 symmetry, or ClusPro was run with two subunits in the multimer docking mode available in the advanced options. The input PDB files consisted of *Mt*SerB2 dimer homology model, *Mt*SerB2 theoretical closed monomer model, or the half of *Mt*SerB2 dimer homology model without hinge-loop (coordinates of chain A ACT2-PSP part (H96-D398) and chain B ACT1 part (A3-R83) of *Mt*SerB2 dimer homology model).

**SAXS-based structure and model evaluation**. Atomic models and crystallographic structures were evaluated by determining the discrepancy ($\chi^2$) between their calculated SAXS curve and the experimental SAXS data using CRYSOL[74] in primus/qt ATSAS 3.0.4 software[75] with default settings (51 points, 15 spherical harmonics, order of Fibonacci grid: 17, solvent density: 0.33 e/$\text{Å}^3$).

**SAXS-based refinement**. *Ma*SerB crystallographic structure[14] (PDB: 3P96) and the best-fitting *Mt*SerB2 tetramer model were refined against the experimental SAXS data using DADIMODO software[76]. Rigid bodies were defined as residues 352–400 of chain A for body1 and residues 352–400 of chain B for body2 for *Ma*SerB, and residues 350–398 of chains A, B, C, D for bodies 1, 2, 3, 4, respectively, for the tetramer model. The evaluation of the discrepancy between the theoretical and experimental SAXS curves was performed with CRYSOL.

**SAXS-based rigid body modelling of *Mt*SerB2 trimers**. Possible *Mt*SerB2 trimer architectures were modelled by rigid body modelling using the ATSAS online version of CORAL software[77] with an overall P3 symmetry. The number of domains was defined as 2, with domain 1 being the ACT1 domain (A7 to E86) and domain 2 being the ACT2-PSP part (H100 to D402). Domain 1 was preceded by a 6-residues-long N-terminal chain and connected to domain 2 by a 13-residues linker. Domain 2 was followed by a 11-residues-long C-terminal chain. Both domains were defined as free.

**L-Ser binding site prediction**. Prior to ligand binding site prediction, *Mt*SerB2 and *Mm*SerB2 dimer homology models, as well as *Ma*SerB crystallographic structure 3P96[14] were prepared using the Protein Preparation Wizard of Maestro 12.9.137 software (Schrodinger). The simulation pH was set to 7.4, and missing side chains were filled in. No restrained minimisation was performed. Hydrogen atoms were subsequently deleted. The prepared structures were submitted to PrankWeb server[78,79] and the prediction was run without the use of conservation.

**Induced-fit docking of L-Ser in *Mt*SerB2 dimer model**. Induced-fit docking of L-Ser was performed in *Mt*SerB2 dimer homology model. The model was first prepared and minimised using the Protein Preparation Wizard of Maestro 11.9.011 software (Schrodinger). The protonation state of the residues was adjusted using Epik at pH 7.4 and the global structure was refined with the OPLS3e force field. L-Ser structure was prepared at pH 7.4 (Epik) by minimisation with OPLS3e force field using LigPrep. The prepared structures were then entered into the Induced Fit Docking protocol. The receptor box centre was defined as the centroid of residues R103, P104, D132, T136 of molecule 2 (chain B) and A3, E33, L34, L35, S53, I89 of molecule 1 (chain A) based on PrankWeb prediction. The box size was set to dock ligands similar in size to L-Ser. Residues within 5.0 Å of ligand poses were refined with an optimisation of the side chains.

**Steady-state kinetics measurements**. Enzyme activity was assayed by free orthophosphate (Pi) determination using a malachite green-based phosphatase assay based on Itaya's colorimetric method[80,81]. The enzyme (1 pmol) was incubated at 37 °C in a total volume of 180 µL containing 25 mM Tris-HCl pH 7.4, 5 mM MgCl2, 1 mM DTT and the desired L-Ser concentration. The reaction was initiated by adding 20 µL of a O-phospho-L-serine (PS) solution at 10× the final well concentration (0–200 mM). After incubation for 10 min at 37 °C, the reaction was stopped by mixing 150 µL of the reaction volume with 50 µL of dye composed of 1.7% ammonium heptamolybdate and 0.22% malachite green in 2 M HCl. The absorbance of the solution was measured at 660 nm. Absorbance due to PS was quantified by replacing the enzyme by the same volume of buffer for each assayed PS concentration, and the obtained value was subtracted from the total absorbance. The activity (released nmol Pi per minute per µg enzyme) was calculated from a calibration curve constructed using dilutions of a phosphate standard solution. All the measurements were made in triplicate.

**Evaluation of the kinetic parameters**. Analysis of kinetic data, curve fitting and statistical analyses were performed using GraphPad Prism 5 (GraphPad Software, La Jolla, California, USA). Values of $K_{M\ (app)}$, $V_{max\ (app)}$ and $k_{cat\ (app)}$ were determined by fitting either Eq. (1) for uncompetitive substrate inhibition for kinetics in the absence of L-Ser, or the compact form of the general modifier Eq. (2)[26] to the initial velocity curves $v$ vs $[S]$

(phosphoserine) in the presence of L-Ser ([I]).

$$v = \frac{V_{max}[S]}{K_M + [S]\left(1 + \frac{[S]}{K_{iS}}\right)} \tag{1}$$

$$\frac{v}{[E]_t} = \frac{k_{cat,app}[S]}{K_{M,app} + [S]} = \frac{k_{cat}\left(1 + \beta\frac{[I]}{\alpha K_i}\right)[S]}{K_M\left(1 + \frac{[I]}{K_i}\right) + [S]\left(1 + \frac{[I]}{\alpha K_i}\right)} \tag{2}$$

Parameters are defined as follows: $v$ is the measured initial velocity (nmol Pi μg$^{-1}$ enzyme min$^{-1}$), $V_{max}$ is the maximum velocity, $[S]$ is the free substrate (O-phospho-L-serine, PS) concentration, $K_M$ is the Michaelis constant, $K_{iS}$ is the dissociation constant of the substrate for the inhibitory site, $[E]_t$ is the total enzyme concentration, $[I]$ is the free inhibitor (L-Ser) concentration, $K_i$ is the inhibitor dissociation constant, $k_{cat}$ is the catalytic constant, α is the reciprocal allosteric coupling constant, β is the factor by which the modifier affects the catalytic constant and the subscript *app* indicates the apparent character of the parameter.

For *Mt*SerB2, the nature of the inhibitory mechanism was determined according to the methodology proposed by Baici[26]. This methodology and the associated terminology are thoroughly described on the website https://www.enzyme-modifier.ch, where is offered a precise and systematic approach to the diagnosis of the kinetic mechanisms of enzymatic inhibition or activation by a modifier compound. As per the approach described above, we were able to identify the nature of the kinetic mechanism of *Mt*SerB2 inhibition by L-Ser by examining the shapes of the plots showing the respective dependencies of $k_{cat,app}$, $1/k_{cat,app}$, $K_{M,app}$, $k_{cat,app}/K_{M,app}$, and $K_{M,app}/k_{cat,app}$ on L-Ser concentration (Eqs. (3–7)). The values of parameters α, β, and $K_i$ were calculated by fitting the developed form of the general modifier Eq. (2) to the $v$ vs $[S]$ curves at fixed L-Ser concentrations.

$$k_{cat,app} = k_{cat}\frac{\left(1 + \beta\frac{[I]}{\alpha K_i}\right)}{\left(1 + \frac{[I]}{\alpha K_i}\right)} \tag{3}$$

$$\frac{1}{k_{cat,app}} = \frac{1}{k_{cat}}\frac{\left(1 + \frac{[I]}{\alpha K_i}\right)}{\left(1 + \beta\frac{[I]}{\alpha K_i}\right)} \tag{4}$$

$$K_{M,app} = K_M\frac{\left(1 + \frac{[I]}{K_i}\right)}{\left(1 + \frac{[I]}{\alpha K_i}\right)} \tag{5}$$

$$\frac{k_{cat,app}}{K_{M,app}} = \frac{k_{cat}}{K_M}\frac{\left(1 + \beta\frac{[I]}{\alpha K_i}\right)}{\left(1 + \frac{[I]}{K_i}\right)} \tag{6}$$

$$\frac{K_{M,app}}{k_{cat,app}} = \frac{K_M}{k_{cat}}\frac{\left(1 + \frac{[I]}{K_i}\right)}{\left(1 + \beta\frac{[I]}{\alpha K_i}\right)} \tag{7}$$

For *Ma*SerB and *Mm*SerB2, the parabolic shape of $1/k_{cat,app}$ vs [L-Ser] combined to the linear dependence of $K_{M,app}/k_{cat,app}$ on [L-Ser] allowed the identification of a mechanism of S-linear I-parabolic noncompetitive inhibition (according to Cleland's nomenclature[82]) that involves the binding of two inhibitor molecules to the enzyme-substrate complex at distinct sites[83]. This mechanism is no longer described by the general modifier equation but is by Eq. (8)[84]. Parameter γ is the allosteric constant

for the binding of the second inhibitor molecule.

$$\frac{v}{[E]_t} = \frac{k_{cat,app}[S]}{K_{M,app} + [S]} = \frac{k_{cat}[S]}{K_M\left(1 + \frac{[I]}{K_i}\right) + [S]\left(1 + \frac{[I]}{\alpha K_i} + \frac{[I]^2}{\alpha\gamma K_{i2}}\right)} \tag{8}$$

On that basis, Eqs. (9–13), derived from Eq. (8), were fitted to the dependencies of $k_{cat,app}$, $1/k_{cat,app}$, $K_{M,app}$, $k_{cat,app}/K_{M,app}$, and $K_{M,app}/k_{cat,app}$ on L-Ser concentration ([I]).

$$k_{cat,app} = k_{cat}\frac{1}{\left(1 + \frac{[I]}{\alpha K_i} + \frac{[I]^2}{\alpha\gamma K_i^2}\right)} \tag{9}$$

$$\frac{1}{k_{cat,app}} = \frac{1}{k_{cat}}\left(1 + \frac{[I]}{\alpha K_i} + \frac{[I]^2}{\alpha\gamma K_i^2}\right) \tag{10}$$

$$K_{M,app} = K_M\frac{\left(1 + \frac{[I]}{K_i}\right)}{\left(1 + \frac{[I]}{\alpha K_i} + \frac{[I]^2}{\alpha\gamma K_i^2}\right)} \tag{11}$$

$$\frac{k_{cat,app}}{K_{M,app}} = \frac{k_{cat}}{K_M}\frac{1}{\left(1 + \frac{[I]}{K_i}\right)} \tag{12}$$

$$\frac{K_{M,app}}{k_{cat,app}} = \frac{K_M}{k_{cat}}\left(1 + \frac{[I]}{K_i}\right) \tag{13}$$

**Statistics and reproducibility**. Enzyme kinetics data are reported as means ± s.d. of three independent enzymatic reactions per condition of substrate concentration, and condition of L-Ser concentration in inhibition kinetics measurements. These mean values were further used in non-linear regression to determine apparent kinetic parameters. The standard errors on these parameters were computed by GraphPad Prism 5.

**Reporting summary**. Further information on research design is available in the Nature Portfolio Reporting Summary linked to this article.

## Data availability
The SAXS data were deposited to SASBDB with accession code SASDRS4 for *Mt*SerB2 dimer, SASDRT4 for *Mt*SerB2 tetramer, SASDRU4 for *Mt*SerB2 trimer, SASDRV4 for *Mm*SerB2 dimer, and SASDRW4 for *Ma*SerB dimer. Uncropped native PAGE gels are shown in Supplementary Figs. 13 and 14. Numerical source data for enzyme kinetics graphs and MALS analyses, as well as MD simulation input files, and initial and final coordinates file, are publicly available on the FigShare repository: https://doi.org/10.6084/m9.figshare.24116571. All other data are available from the corresponding author on reasonable request.

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

## Acknowledgements
The pAVA0421 plasmids encoding *Mt*SerB2, *Ma*SerB and *Mm*SerB2 were kindly provided by the Seattle Structural Genomics Center for Infectious Disease (www.SSGCID.org) which is supported by Federal Contract No. 75N93022C00036 from the National Institute of Allergy and Infectious Diseases, National Institutes of Health, Department of Health and Human Services. SAXS experiments have been supported by iNEXT-Discovery, grant number 871037, funded by the Horizon 2020 program of the European Commission. We acknowledge SOLEIL for the provision of synchrotron radiation facilities and we would like to thank Dr. J. Pérez and Dr. A. Thureau for assistance in using beamline SWING (proposal number 20210718). We thank Dr. A. Demelenne (LAM, CIRM, ULiège) for her precious help on the SEC-UV-MALS equipment. The Fonds de la Recherche Scientifique (F.R.S.-FNRS, Belgium) is acknowledged for E.P. Research Fellow - ASP grant and for the MALS detector.

## Author contributions
E.P. and F.D.P. designed the experiments, prepared the samples, acquired, processed and analysed the data, and co-wrote the manuscript. J.W. conceived and supervised the project. M.F. provided access to SEC-UV-MALS and MP equipment and discussed the results. All authors reviewed the manuscript and approved the final draft for submission.

## Competing interests
The authors declare no competing interests.
