## [Peer Review File · Communications Biology]

Reviewers' comments:

Reviewer #1 (Remarks to the Author):

This manuscript by Pierson et. al. describes an extensive study of the conformational states of M. tuberculosis SerB2 in the presence and absence of L-serine and presents a model of how serine inhibits enzyme activity. The manuscript is well written and the experiments are rigorously performed. The end product is an excellent study of the allosteric control mechanism of this enzyme. The results are novel and will be of interest to a wide community. However, there are some questions that need to be addressed as well as some suggestions to improve the manuscript.

The mechanism depicted in Fig. 3c invokes a model where there are 2 distinct substrate binding sites. Why? What is the evidence for 2 sites? A more general model for substrate inhibition is a model where there is a single site but as the substrate concentration increases it can force unproductive binding orientations that compete with the productive orientation.

The authors state that there is no shift in the plot in Fig S8 for the delta ACT1 mutant. However, there does appear to be a slight shift. Could it simply be that the mutant is less sensitive to L-ser? What is the E:ser ratio here as compared to Fig 7b?

In Fig 5e, the numbering does not correspond to the residue numbers that were mutated in the Mtb enzyme. This is likely because the numbering of the two enzymes differ slightly in this region. But, the number differences can be initially confusing, so this should be stated in the text that the residues actually do align.

On line 196, which species does the "analyzed species" refer to?

In figure 6d, I do not understand the need to invoke a "cutting" of the hinge region and a reconnection. Surely, this does not represent a viable in vivo mechanism for formation of a tetramer. If this is purely an in silico step needed to form this particular tetramer, it should be clearly stated as such and does not imply an actual mechanism.

In fig 6, the best fit tetramer is purported to be a tetramer of closed monomers. As this being the only data that supports this, it should be made clearer why or how a tetramer of open monomers is ruled out. Perhaps by showing the generated SAX fit for comparison. As it is, it is not clear how the evidence supports this arrangement as opposed to a tetramer of open monomers (dimer of open dimers). The first point given at lines 228 and 229 would also hold for the open monomer. And, the rationale for the second point involving the helices is based on the human counterpart. I get the logic, as far as it goes, but the evidence is scant. Comparing the Mtb model directly, and in detail, with the avian structure might help solidify this conjecture.

Why wasn't mass photometry used to analyze the putative trimer formation? It seems somewhat inconsistent to use it earlier and then not here.

Some of the figures are problematic in that the inclusion of many panels within the same figure often does not provide sufficient detail to enable the reader to clearly see the aspects referred to in the figure. In addition, the resolution of the figures that were included with the submission is not of sufficient resolution to allow them to be expanded with clarity. These include:

Insert to figure 4d

Insert to Fig 6e

Figure 6 g and h

Figure 6i doesn't clearly show how the helix blocks the active site.

Figure 7d, the putative pocket is not clearly shown in this rendition. Its location, but not its form is all

that can be discerned.

Figure 7 e would greatly benefit from providing a stereo view

Other problems with the figures are as follows:

Fig 5d, the colors and line types are not sufficiently contrasting to clearly distinguish them. Same for 7f.

It is not clear which model in Fig 6d corresponds to the best fit structure in 6f.

Fig 7i is referred to in the text (line 232) but not shown in the figure.

Does line 198 which says 7d actually mean 6d?

In Fig 8, what is $\text{HMx}(\text{Sp} > \text{Ca})\text{I}$ and why is it there?

Fig S10, the slope replot clearly contains an outlier that should be eliminated so the fit to the rest of the data is clearer.

Other:

The wording at lines 284-285 is missing something. Should it be "stabilising of the ACT1.."

Reviewer #2 (Remarks to the Author):

The study by Pierson et al, describes the kinetic and biophysical analysis of an interesting enzyme phosphoserine phosphatase SerB2 from *M. tuberculosis*. This is a very nice piece of work with detailed kinetic and SAXS analysis. Overall, the applied methods are sound and the conclusions regarding the formation of different oligomers are reasonable. However, there are some aspects which need more clarification. The speculation for the dimer being domain-swapped and the tetramer not being domain-swapped mainly comes from homology modeling and SAXS analysis (to some extent also from mass photometry studies). Since there are key differences in the hinge region connecting the two ACT domains in MtSerB2 when compared to its close homologs and the mutation in the hinge region does not significantly affect the formation of tetramer, it would be better to add more direct evidence for these states. For example, a thermal unfolding experiment and/or chemical crosslinking-mass spectrometry experiment can add solid proof of the claims presented.

The kinetic studies carried out are excellent. It is important to describe these more in the results section. It is impossible to understand the derived conclusions without reading several referred articles. Therefore, it is suggested to expand this section for better clarity and understanding of the general reader. Two types of inhibitions have been reported here – but the relation between them is not at all discussed. How are the substrate and product inhibition related? Is the allosteric site for the substrate and the product same? In Grant's paper (Biochemistry), it is also described that the product inhibition happens only in the presence of phosphate ions. But that doesn't seem to be the case as reported here – or at least it has not been discussed here in detail.

More detailed comments are below. There may be some overlapping comments.

1. Line 109: No change in oligomeric equilibrium was observed...

But Fig. 2D lane 2 has no dimer population – is this a concentration effect?

Dimer population observed in tetramer peak is it domain-swapped or closed?

Any molecular weight markers used for this?

Full native PAGE for Fig 2d and 3a can be added to the supplementary information.

2. Line 111: two weeks and three months' time...

Can include that the 'population of dimer and tetramer after two weeks and three months' time compared with the population in freshly thawed protein....'

Color coding in Fig. 2e: the labeling can be in the same order for all peaks (Fresh, 2-weeks and 3 months) It is a little challenging to compare the values in figure.

Is the peak height difference due to difference in the concentration of protein used?

Also, has the enzymatic activity checked after two weeks and three months, especially for the dimer pool?

The calculated molecular weight by mass photometry is lower than expected tetramer and dimer Mwt, especially for the enzyme after three months. Any degradation or stability issues observed?

3. Line 119: Fig. 3a full native page in supplementary information should be added

4. Line 120: The enzyme kinetic section should be expanded and explained in more detail considering the special nature of this enzyme. If this is not new information a reference for previous studies should be added and stated clearly if these experiments were only repeated for reproducibility.

It is stated as uncompetitive substrate inhibition – are the additional substrate binding sites known? How was this uncompetitive inhibition concluded?

There is clearly inhibition of reaction when using higher substrate concentrations according to Fig. 3b and e. Is this because of substrate binding at a different allosteric site or the product formed being binding at different site? Considering that Ser is also shown to bind allosterically, is the observed inhibition, substrate inhibition or product inhibition? How can that be differentiated?

The inhibiting substrate concentrations are also very high- are they physiologically relevant substrate concentrations?

K_{si} – 2.7 mM. Not much sign of inhibition up to 2.0 mM in Fig 8. a

Why is the K_M referred as substrate dissociation constant – this is probably true only in some special cases. Is this known for SerB2? If yes, provide appropriate reference. Or it is better to simply mention K_M as Michaelis constant.

5. Line 126: The dimer percentages slightly different from that in Fig. 2d.

6. Line 143: The domain swapping in MtSerB2 is assumed mainly from homology modeling and fit to SAXS curves. Since the best fitting was achieved after some 'relaxation', has the authors also checked non-domain swapped dimers using the same procedure to see if the non-domain swapped model fits well to the SAXS data?

Since Ma and MtSerB2 have distinct properties despite their high sequence similarity, it would be good to have additional experimental evidence for the presence of domain swapping in MtSerB2, for example thermal unfolding and/or chemical crosslinking mass spectrometry experiments.

While it is clear that the studies on MtSerB2 delta Act1 variant confirm that this domain is important for dimer formation, it is not clear how it confirms domain-swapping.

There is no mention of any crystallization attempts for this enzyme. Has this been tried and did not crystallize?

7. Line 193: monodisperse elution peak – is it not that the tetramer is in equilibrium with some population of dimer?

8. Line 217: The organization of the domains in the final selected model should be explained better. The final selected model from SAXS studies also seems to be a dimer of dimer (D2 symmetry). To me it looks like that the two dimers are interacting through the C-terminal catalytic (PSP) domain with the ACT domains pointing outwards. Is that correct? i.e. similar to model 1 in Fig. 6d but interaction through the PSB domain. This particular arrangement with the interaction through PSP domain is not depicted in Fig. 6d. It would be good to add this.

Also, in Fig. 6f, it would be worthwhile to show the relative position of the domains to have better clarity on how the two dimers are interacting and/or color one monomer with different domain colors as in Fig. 1.

Has a similar model with domain swapping also been checked for fitting against SAXS data? How would that fit?

The tetramer model is better depicted in Fig. 9, a similar figure can be included in Fig. 6

9. Any relation between trimer and tetramer formation?

10. Is the substrate and product binding allosteric site the same? Is the substrate inhibition shown in Fig. 1 different from product inhibition? Fig. 8a shows normal MM kinetic behavior up to 2.0 mM substrate concentration. In Fig. 1 up to 20 mM substrate has been used. Why?

11. Line 305: partial (hyperbolic) mixed, predominantly competitive (specific) inhibitor – Explain in more detail. The information is quite scattered now in Figs and methods section. It is important to explain how this conclusion was reached in the results section – from a general reader point of view. Effect of phosphate ion for the inhibition not mentioned – discussed in Grant (Biochemistry).

12. Can the SAXS data differentiate between domain-swapped and closed dimer/tetramer? – probably not. Whether it is closed or domain-swapped dimer/tetramer, the interaction between Act1 and Act2 are the same. Therefore, for the final best fitting models, the corresponding domain-swapped or closed oligomer fitting should also be tested.

13. Line 871 Fig. 1 legend: ACT domains not yellow and purple but yellow and red

14. Line 872 Fig. 1 legend: The Mg²⁺ ion in green is not visible. The color of Mg²⁺ ions should be changed or highlighted for better visibility.

Reviewer #3 (Remarks to the Author):

Pierson et al provide outstanding evidence that SerB2 from *M. tuberculosis* uses the morpheein model of allostery to control catalysis and that pharmacological stabilization of one or more alternate assemblies is a promising approach to new therapies. The work combines a variety of methods, both experimental and computational, directed mutagenesis, as well as sophisticated modelling to fit SAXS data, to support the existence of a flexible monomer, a dimer, at least one tetrameric assembly, and an unexpected trimeric assembly that is the L-Ser-stabilized allosterically inhibited form. Suggestion that alternate assemblies may correlate with SerB2 moonlighting functions further defines the morpheein character of SerB2. So, why does the title use the term morpheein-like? Strongly suggest removing the “-like” from the title. SerB2 uses the morpheein model of allostery. The only significant concern is the use of the term induce. The morpheein model of allostery is not an induced fit model. The ligand does not bind to one form and induce a conformational change. The protein exists in an equilibrium of assemblies wherein the ligand binding site is on one assembly and not on the others. The authors say the different assemblies coexist, interconverting through a

conformationally flexible monomer. The mechanism is conformational selection. Note however the description lines 283-286. Does L-Ser bind to the dimer? Or, does L-Ser bind to the trimer and pull the equilibrium in that direction? See also lines 361-365 and lines 1043-1044.

Minor points, suggestions, and corrections:

1. The authors might point out that the conformational change in the monomer (open vs closed) is a repositioning of folded domains, a hinge motion, and does not include any refolding (as is found in metamorphic proteins).
 2. The authors might point out that control of catalysis depends on quaternary structure changes that control active site access. This is a common theme among morphoheins.
 3. Proteins with amino acid substitutions should be called variants. They are not mutants (e.g. bottom of p. 7).
 4. Since it takes only a few specific amino acid substitutions to cause SerB2 to lose the ability to populate alternate assemblies, how might this relate to the eventual development of resistance to the promised therapeutic?
 5. Fig 1a – legend – the N-terminal ACT domains are not yellow and purple. They are yellow and red. Reviewer cannot see the green sphere.
 6. Fig 2d – lanes 1 and 2 suggest the presence of two conformationally distinct tetramers. Although it is not possible to investigate this using SEC, one may be able to separate these forms using IEC. This is a suggestion for future study.
 7. Line 887 – change to “at a 90° angle”.
 8. Note that Fig 3a shows a clean single tetramer band. How is the sample different from Fig 2d?
 9. Line 901 – replace flash-freezed with flash-frozen.
 10. Line 950 – what is a low-conversation score?
 11. Where possible, make the text in the figures larger.
 12. Fig 7d requires higher quality images.
 13. Fig 8 – Remove the artifact along the right edge of the figure.
- E.K. Jaffe

Reviewer #1 (Remarks to the Author):

This manuscript by Pierson et. al. describes an extensive study of the conformational states of M. tuberculosis SerB2 in the presence and absence of L-serine and presents a model of how serine inhibits enzyme activity. The manuscript is well written and the experiments are rigorously performed. The end product is an excellent study of the allosteric control mechanism of this enzyme. The results are novel and will be of interest to a wide community. However, there are some questions that need to be addressed as well as some suggestions to improve the manuscript.

Author response:

We would like to thank the reviewer for their flattering comments and acknowledgement of the impactful character of our work. Their constructive remarks helped us clarify several points in the text and further improve our manuscript.

The mechanism depicted in Fig. 3c invokes a model where there are 2 distinct substrate binding sites. Why? What is the evidence for 2 sites? A more general model for substrate inhibition is a model where there is a single site but as the substrate concentration increases it can force unproductive binding orientations that compete with the productive orientation.

The substrate inhibition mechanism we propose here is the classical model developed by Haldane in 1930, and is the one most commonly used to describe this phenomenon in enzymes. It involves a ternary complex where two molecules of substrate are bound to the enzyme. The ternary complex either can produce the product at a reduced rate or is unproductive, as is the case here since the initial velocity approaches zero at high substrate concentration (**Figs 3b and 3e**). This substrate inhibition mechanism with an unproductive ternary complex is depicted in **Fig 3c**. **Eq1** was derived from that mechanism using the rapid equilibrium assumption and fits the data well.

However, the formation of the ternary complex can either be interpreted as:

(1) the binding of the first substrate molecule at the active site and the binding of the second substrate molecule at an allosteric site, or (2) the binding of two substrate molecules at the active site in an unproductive manner, as is suggested by the reviewer.

In both cases, the form of **Eq1** remains the same. Discriminating between the two options in the case of *MtSerB2* and the other mycobacterial PSPs would require further mechanistic studies. Nevertheless, **we changed the notation designating the ternary complex in Fig 3c to "ES²" instead of "SES"** so that the notation encompasses these two possibilities and does not imply the demonstrated existence of two distinct binding sites.

The authors state that there is no shift in the plot in Fig S8 for the delta ACT1 mutant. However, there does appear to be a slight shift. Could it simply be that the mutant is less sensitive to L-ser? What is the E:ser ratio here as compared to Fig 7b?

Please note that **Fig S8** is now **Fig S10**. The experiment shown in **Fig S10** was carried out with a E:Ser ratio of 1:1000. The monomer peaks appearing in the SEC chromatograms of **Fig S10** for Δ ACT1 variant in presence (red) and absence of L-Ser (black) are indeed not perfectly aligned. In the presence of L-Ser, the monomer elutes about 9 seconds earlier (18.85 min vs 19.00 min). We recognize that binding of L-Ser to the artificially truncated variant of *MtSerB2* could indeed occur

to some extent, perhaps at the surface of the ACT2 domain. This could result in a small change in the hydrodynamic radius which would explain the slight 9 seconds shift observed in **Fig S10**.

However, although this assumption may be valid, this event is unlikely related to a change at the quaternary level, as we state in the text as follows (**lines 286-288**):

“Additionally, while the variants altered in their ability to tetramerize still underwent the transition, the oligomeric state of variant MtSerB2ΔACT1 was left unchanged in the presence of L-Ser (...).”

Complementary experiments such as ITC could demonstrate an eventual interaction between L-Ser and ΔACT1 variant.

In Fig 5e, the numbering does not correspond to the residue numbers that were mutated in the Mtb enzyme. This is likely because the numbering of the two enzymes differ slightly in this region. But, the number differences can be initially confusing, so this should be stated in the text that the residues actually do align.

Yes, MaSerB indeed has 2 more residues at the beginning of its sequence. We stated in **Fig 5e** legend (**lines 1021-1025**) that the residues are homologous:

“(...) residues A150, D151, E152 and T156 (homologous to key residues C148, V149, G150 and I154 for tetramerization in MtSerB2) (...).”

For more clarity, **we added the following sentence** to the legend (**lines 1024-1025**):

“The difference in numbering comes from the fact that MaSerB bears two more residues than MtSerB2 at the N-term.”

We also **added a more explicit indication** directly in **Fig 5e** with the sentence “These residues align with C148, V149, G150 and I154 in MtSerB2”.

On line 196, which species does the “analyzed species” refer to?

(Line 223 [196 in previous version]): “analysed” was replaced by “tetrameric”

In figure 6d, I do not understand the need to invoke a “cutting” of the hinge region and a reconnection. Surely, this does not represent a viable in vivo mechanism for formation of a tetramer. If this is purely an in silico step needed to form this particular tetramer, it should be clearly stated as such and does not imply an actual mechanism.

In **Fig 6d**, we explain how we modelled different models of plausible MtSerB2 tetramers from the structural data in our possession. The steps depicted are indeed purely *in silico* steps. We started out from the atomic coordinates (.pdb file) of MtSerB2 domain-swapped dimer homology model and used the already existing hinge-loop of chain B. The atoms corresponding to the PSP-ACT2 part of chain B and to the ACT1+hinge-loop part of chain A were deleted from the pdb file. Then the bond between the hinge-loop of chain B (grey) and the ACT2 domain of chain A (pink) was re-created *in silico*. The resulting molecule was submitted to a minimization of energy and Molecular Dynamics simulation to model a plausible closed monomer that could be docked with three other copies of itself to model a tetramer.

This matter was **clarified by making precisions** in the legend of **Fig 6d**:

(line 1039) “Strategy used for the modelling of MtSerB2”

became

“Strategy used for the in silico modelling of MtSerB2”

and

(line 1041) *“The procedures for constructing three different types of tetrameric architectures”*

became

“The in silico modelling steps for constructing three different types of tetrameric architectures”

In fig 6, the best fit tetramer is purported to be a tetramer of closed monomers. As this being the only data that supports this, it should be made clearer why or how a tetramer of open monomers is ruled out. Perhaps by showing the generated SAX fit for comparison. As it is, it is not clear how the evidence supports this arrangement as opposed to a tetramer of open monomers (dimer of open dimers). The first point given at lines 228 and 229 would also hold for the open monomer. And, the rationale for the second point involving the helices is based on the human counterpart. I get the logic, as far as it goes, but the evidence is scant. Comparing the Mtb model directly, and in detail, with the avian structure might help solidify this conjecture.

As we understand it, the reviewer asks here why and how we support the idea that the tetramer is composed of 4 closed monomers, as shown in **Figure 6**, rather than presenting an identical overall architecture but with exchange of ACT1 domains between neighbouring subunits (i.e. the same tetramer model but with domain-swapping, where the monomers would therefore be in an ‘open’ conformation).

Based on SAXS data only, we could not rule out the possibility that there is ACT1 domain-swapping in the tetramer model presented in **Fig. 6**. The low spatial resolution and information content of SAXS data would not allow the detection of a conformational change at the hinge-loop level because the overall shape of the protein model (relative positions of the domains) remains identical whether or not there is ACT1 domain-swapping between the subunits. Based on SAXS data only, both architectures are equally plausible.

However, throughout the manuscript, we present additional observations that further support the model obtained following SAXS-based screening. These are the following:

First, as shown in **Fig. 2d** and discussed at **lines 109-110 and 241-243**, we draw the reader’s attention to the observation that concentrating the dimeric fraction up to 25.5 mg/mL does not result in the formation of the tetramer. If the tetramer was a dimer of domain-swapped dimers, we would expect that such a high concentration favours interactions between dimers and hence forces tetramer appearance in solution. However, it is not the case. Conversely, the conversion of a domain-swapped dimer to a tetramer made of four closed monomers involves a dissociation step of the dimer into its monomeric constituents. In such a set-up, concentrating the dimer would not result in formation of the tetramer. That is what is observed.

Secondly, the ‘four closed monomers’ architecture accounts for the observations we made regarding the relative stabilities of the tetramer and the dimer, and their rate of interconversion. As highlighted by mass photometry (**Fig 2e**), the tetramer quickly converts to dimer while the dimer needs three months to reform a population of tetramer. As discussed **from line 411** and depicted in **Fig 9e**, this is probably because the separation of four stable closed monomers (that are in dynamic equilibrium with their open form according to domain-swapping principles - see

ref 33) is less energetically costly than the separation of a more stable intertwined dimer into two open monomers of high energy.

Finally, the 'four closed monomers' architecture can also explain why *MtSerB2* forms a small population of tetramer in addition to the more stable dominant dimeric population: based on **Fig 9e**, right after ribosomal translation, the freshly translated monomeric chain would probably adopt its most stable conformation: a closed conformation (with ACT1-ACT2 intramolecular interactions). We discuss this **from line 416**. The formation of the dimer is favoured by thermodynamic factors: it is the most stable oligomer but its formation requires overcoming an energy barrier imposed by the opening of the monomer. Conversely, the formation of the tetramer (less stable than the dimer, but more than an isolated monomer) would be kinetically favoured, as this process only needs four monomers to interact together without energetically costly conformation changes.

Additionally, in the domain-swapped homology model of *MtSerB2* dimer, the hinge loop adopts an extended conformation to link R83 to H96, whose C α are separated by 29.9Å. In the tetramer model that we propose, the distance between R83 and H96 of adjacent subunits is 39.9Å. In that configuration, it would require the hinge-loop to be completely extended to connect these two anchoring points, which would be strongly entropically unfavourable.

Although we understand that higher resolution structural techniques would be needed to univocally validate our model, these observations converge and support that *MtSerB2* tetramer is most likely composed of four closed monomers.

Why wasn't mass photometry used to analyze the putative trimer formation? It seems somewhat inconsistent to use it earlier and then not here.

While not mentioned in the manuscript, we also attempted to analyse trimer formation using mass photometry but trimer could not be detected using this technique. The extremely diluted conditions required by MP probably do not favour the association of the monomers in solution to form the trimer, as opposed to the concentration conditions of SEC experiments.

Some of the figures are problematic in that the inclusion of many panels within the same figure often does not provide sufficient detail to enable the reader to clearly see the aspects referred to in the figure. In addition, the resolution of the figures that were included with the submission is not of sufficient resolution to allow them to be expanded with clarity. These include:

Insert to figure 4d

Insert to Fig 6e

Figure 6 g and h

We are aware that the figures contain numerous panels. We have done our best to comply with the journal's recommendations (around ten display items) while illustrating our points as detailed as possible. The figures were directly included in the word document in .png format for the reviewing and it indeed affected their resolution. We can assure that the figures were however prepared at a high resolution mostly using vector graphics. We made sure they could be expanded with clarity using a pdf reader software and the largest zoom setting. We believe that they will be processed at full quality by the editing services for the online/final version of the manuscript when we provide the original files along with our submission.

Figure 6i doesn't clearly show how the helix blocks the active site.

Figure 7d, the putative pocket is not clearly shown in this rendition. Its location, but not its form is all that can be discerned.

Figure 7 e would greatly benefit from providing a stereo view

We thank the reviewer for these three remarks. Unfortunately, a two-dimensional figure will never faithfully convey the complexity of a 3D model. To help solve this difficulty, we are making the final 3D models available in SASBDB under the accession codes SASDRS4, SASDRT4, SASDRU4, SASDRV4 and SASDRW4. As referred to in **Fig. 6j** (new version of **Fig. 6i**) legend, we also invite the reviewer to see ref 25 (Haufroid, Mirgoux *et al.*) and PDB structure 6HYY for more detailed figures on the dynamic mechanism of active site shielding by the helix from the C1-cap. We are aware that the figures included in the version of the paper intended for reviewing were not of sufficient resolution. In the final high-resolution version of **Fig. 7**, the shape of the pocket can be discerned by zooming-in (computer version). We also added an insert that gives a closer view of the pocket on **Fig. 7e**. As for **Fig. 7e** itself, we believe that the essential points have been included: it clearly shows that L-serine interacts with 4 residues (R103, T136, E33 and L34). The following interactions are illustrated: ionic interactions between the guanidinium group of R103 and the carboxylate of L-Ser, and between the carboxylate of E33 and the ammonium of L-Ser, and H-bonds between the carboxylate of E33 and the ammonium of L-Ser, the backbone carbonyl of L34 and the hydroxyl of L-Ser, the hydroxyl of T136 and the hydroxyl of L-Ser, and the hydroxyl of T136 and the carboxylate of L-Ser.

Other problems with the figures are as follows:

Fig 5d, the colors and line types are not sufficiently contrasting to clearly distinguish them. Same for 7f.

The **contrast of the colors and line types have been modified** in order to make **Fig 5d** and **Fig 7f** easier to interpret.

It is not clear which model in Fig 6d corresponds to the best fit structure in 6f.

Please note that the left panel of **Fig. 6f** (previous version) is now **Fig. 6e** (new version).

Figure 6d does not attempt to represent every three-dimensional model that can be generated by protein-protein docking by a 2D diagram. Given the complexity of symmetry, perspective and the number of different structures generated by docking, this is not feasible. However, we recognize that representing only one possible architecture for each approach (1), (2) and (3) generated a certain amount of confusion in relation to obtaining the best-fit structure represented in **figure 6e** (new version). We have therefore **added other representations of possible architectures, plus signs and suspension marks** to **Fig. 6d** in order to better illustrate the diversity of structures that were obtained through the *in silico* modelling process shown there.

Extracts from Fig 6d:

PREVIOUS VERSION

NEW VERSION

Fig 7i is referred to in the text (line 232) but not shown in the figure.

It is a typo. We were in fact referring to **Fig. 6i**. Please note that **Fig 6i** (previous version) is now **Fig. 6j** (new version).

Changed in the text by “Fig 6j”.

Does line 198 which says 7d actually mean 6d?

Yes, it is a typo. Changed in the text by “Fig. 6d”.

In Fig 8, what is HMx(Sp>Ca)I and why is it there?

HMx(Sp>Ca)I is the inhibition type and is the short writing form for “Hyperbolic Mixed predominantly Specific Inhibition” according to A. Baici’s nomenclature (<https://www.enzyme-modifier.ch>). This inhibition type was diagnosed from the shapes of the plots of apparent k_{cat} , K_M , k_{cat}/K_M and their multiplicative inverses vs [L-Ser] concentration as per Baici’s methodology. As also suggested by another reviewer, **we expanded our kinetics section (lines 342-363, 367-371, 372-379 and 695-701)** to make it more accessible from a general reader point of view. The section now explains the mechanism diagnosis procedure and its origin in detail. It is now easier

to understand the nomenclature used in **Fig 8** and why it is written next to these particular replots of apparent kinetics constants vs [L-Ser]. Please note that the lettering of the panels in **Fig 8** have been changed: **Fig 8b→8c; Fig 8c→8b**. The **legend has been modified consequently** and **a line about the diagnostic methodology** has been added.

Detailed modifications:

Results section, PREVIOUS VERSION : *“from which apparent k_{cat} and K_M values could be derived by non-linear regression using the condensed form of the general modifier equation (2). The kinetic mechanism was diagnosed by evaluating the dependences of apparent k_{cat} , K_M , and specificity constant (k_{cat}/K_M) on L-Ser concentration. Through this method, we identified L-Ser as a partial (hyperbolic) mixed, predominantly competitive (specific) inhibitor of MtSerB2 (Fig. 8b and 8c), which agrees with Grant’s analysis . The slight partial character was confirmed by the plot of initial velocities versus L-Ser concentration showing a plateau (Fig. 8d).”*

changed to

Results section, NEW VERSION (**lines 342-363**): *“As initial rates decreased with increasing [L-Ser], we could conclude that L-Ser inhibited MtSerB2 phosphatase activity. To identify the kinetic mechanism of inhibition, we followed the systematic approach proposed by Baici²⁶ (thoroughly described on the website <https://www.enzyme-modifier.ch>). First, apparent k_{cat} and K_M values could be determined by fitting the condensed form of the general modifier equation (2) to the v vs $[S]$ data shown in Fig. 8a. Next, the inhibition mechanism was diagnosed by replotting the apparent k_{cat} , K_M , specificity constant (k_{cat}/K_M), and their respective multiplicative inverses versus [L-Ser] and examining the shapes of the plots (Fig. 8b). As described on Baici’s website, each kinetic mechanism gives a unique combination of replots. The shape of each replot is designated by a letter, depending on how each parameter behaves when [inhibitor] increases (Supplementary Table 1): independent (A, D, H, L, O); increases hyperbolically (B, E, I, M, P); decreases hyperbolically (C, F, J, N, Q) or increase linearly (G, K, R). For MtSerB2, the shape of the parameters dependencies on [L-Ser] were found to match the combination C-E-I-N-P (Fig. 8b). This combination corresponds to a hyperbolic mixed, predominantly specific inhibition (‘HMx(Sp>Ca)’) according to Baici nomenclature) mechanism, which, in more common terms, refers to a mechanism of partial, predominantly competitive inhibition. In this mechanism (Fig. 8c), L-Ser can bind to the enzyme-substrate complex but with a lower affinity ($\alpha > 1$) than to the free enzyme (predominantly competitive character). The ternary complex formed between the enzyme, the substrate and L-Ser is still able to form product, but at a slower rate ($\beta < 1$, partial character). This behaviour agrees with Grant’s conclusions¹⁹. The partial nature of the inhibition was slightly ambiguous in view of the quasi-linear dependence of apparent $1/k_{cat}$ on [L-Ser] but was confirmed by the plot of initial velocities versus L-Ser concentration showing a plateau rather than reaching zero (Fig. 8d).”*

Results section, PREVIOUS VERSION : *“These results are in line with our trimer models, whose interfaces formed at the PSP domains could restrict access to the active site.”*

changed to

Results section, NEW VERSION (**lines 367-371**): *“These results are in line with the formation of a weakly active trimer upon disruption of the dimeric assembly. L-Ser would interact more easily with the free dimer but could also disrupt MtSerB2-phosphoserine complex. Our trimer*

models, whose interfaces formed at the PSP domains could restrict but not totally prevent access to the active site, explain the remaining low activity.”

Results section, PREVIOUS VERSION : *“In addition, using the same methodology, we diagnosed L-Ser as a total parabolic mixed, predominantly uncompetitive inhibitor of MaSerB and MmSerB2 (Supplementary Fig. 10).”*

changed to

Results section, NEW VERSION (lines 372-379): *“In addition, using the same methodology, we diagnosed L-Ser as a total parabolic mixed, predominantly uncompetitive (or ‘S-linear I-parabolic noncompetitive’) inhibitor of MaSerB and MmSerB2 (Supplementary Fig. 12). In this mechanism, L-Ser can bind both the free enzyme and the enzyme-substrate complex (with more affinity), and a second L-Ser molecule then binds the enzyme-substrate-L-Ser complex to totally inhibit enzyme activity. On the basis of MaSerB-L-Ser cocrystal structures (PDB: 5JLR27 and 5JLP28), we hypothesize that L-Ser would interact with residues D17 and I126 at the ACT1-ACT2 domain interface (Fig. 7d) and that a second binding site leading to total inhibition would appear after a conformational change triggered by substrate binding.”*

Methods section, PREVIOUS version: *“For MtSerB2, the nature of the inhibitory mechanism was determined by assessing the shapes of the 623 dependencies of (...)”*

changed to

Methods section, NEW VERSION (lines 695-701): *“For MtSerB2, the nature of the inhibitory mechanism was determined according to the methodology proposed by Baici²⁶. This methodology and the associated terminology are thoroughly described on the website <https://www.enzyme-modifier.ch>, where is offered a precise and systematic approach to the diagnosis of the kinetic mechanisms of enzymatic inhibition or activation by a modifier compound. As per the approach described above, we were able to identify the nature of the kinetic mechanism of MtSerB2 inhibition by L-Ser by examining the shapes of the plots showing the respective dependencies of (...)”*

Fig S10, the slope replot clearly contains an outlier that should be eliminated so the fit to the rest of the data is clearer.

Corrected as per the reviewer’s suggestion. Please note that **Fig. S10** is now **Fig. S12**. The 275 μM L-Ser point indeed had a large error on $(K_M/k_{\text{cat}})_{\text{app}}$ vs [L-Ser] in **Fig S12b** (left). The large error came from an inaccuracy in determining apparent K_M by non-linear regression using **equation 8** from the data of **Fig. S12a** (left). As this point is linked with the 4 other replots of **Fig S12b** (left), we chose to eliminate it from all data sets. Performing the non-linear regression again didn’t change the shape of the replots and the diagnosed mechanism therefore remains the same.

Other:

The wording at lines 284-285 is missing something. Should it be “stabilising of the ACT1..”

Indeed. **Changed as follows:**

(lines 324-325) *“ by disrupting the stabilising the ACT1-ACT2 intermolecular interface.”*

Reviewer #2 (Remarks to the Author):

The study by Pierson et al, describes the kinetic and biophysical analysis of an interesting enzyme phosphoserine phosphatase SerB2 from *M. tuberculosis*. This is a very nice piece of work with detailed kinetic and SAXS analysis. Overall, the applied methods are sound and the conclusions regarding the formation of different oligomers are reasonable. However, there are some aspects which need more clarification. The speculation for the dimer being domain-swapped and the tetramer not being domain-swapped mainly comes from homology modeling and SAXS analysis (to some extent also from mass photometry studies). Since there are key differences in the hinge region connecting the two ACT domains in MtSerB2 when compared to its close homologs and the mutation in the hinge region does not significantly affect the formation of tetramer, it would be better to add more direct evidence for these states. For example, a thermal unfolding experiment and/or chemical crosslinking-mass spectrometry experiment can add solid proof of the claims presented.

The kinetic studies carried out are excellent. It is important to describe these more in the results section. It is impossible to understand the derived conclusions without reading several referred articles. Therefore, it is suggested to expand this section for better clarity and understanding of the general reader. Two types of inhibitions have been reported here – but the relation between them is not at all discussed. How are the substrate and product inhibition related? Is the allosteric site for the substrate and the product same? In Grant's paper (Biochemistry), it is also described that the product inhibition happens only in the presence of phosphate ions. But that doesn't seem to be the case as reported here – or at least it has not been discussed here in detail.

More detailed comments are below. There may be some overlapping comments.

Author response:

We would like to thank the reviewer for the very positive evaluation of our work. The reviewer pointed out certain aspects that needed further clarification and this enabled us to rework several parts of the manuscript in order to improve it. In particular, we have been able to provide additional arguments as to the domain-swapped architecture of the dimer, further detail the structure of the best tetramer model obtained, and develop our kinetics section in order to make it accessible from the general reader's point of view. The aspects raised above by the reviewer are covered in detail in the questions below. We will therefore answer them point by point.

1. Line 109: No change in oligomeric equilibrium was observed...

But Fig. 2D lane 2 has no dimer population – is this a concentration effect?

In **Fig. 2d**, it can indeed be observed that lane 2 of the native PAGE gel has a more faded lower band. We believe that it is most likely an artifact observation due to electrophoresis conditions. We do not associate the fading of the lower band with concentration effects since lane 4 unambiguously reveals that the dimeric population is unaffected by concentration.

Dimer population observed in tetramer peak is it domain-swapped or closed?

According to our model, the trace amount of dimer population in the tetramer peak is most likely to be domain-swapped. We have no evidence that a dimer of closed monomers could exist in solution but think that such an architecture would be unlikely due to the absence of the most stabilizing interactions at the dimer interface. **We invite the reviewer to refer to the answer we provide to remark #6 below.**

Any molecular weight markers used for this?

In our protocol of native PAGE, the loading buffer does not contain any additive to give protein a uniform net charge. The net charge of the protein is only dependent on the pH of work. The electrophoretic mobility therefore depends on size but also on the isoelectric point of the analyte. In denaturing conditions, such as in SDS-PAGE, SDS is added so that the mass-to-charge ratio of each analyte is identical, which allows the use of a molecular marker for semi-quantitative measurement of molecular weight since migration is only dictated by the size of the protein. In native PAGE conditions, the use of a molecular weight marker could be considered biased because each protein has a particular mass-to-charge ratio, and hence migrate according to size and net charge. For this reason, and because we already knew the composition of our samples (dimer and/or tetramer) thanks to SEC, we chose not to use molecular weight markers in our native PAGE experiments.

Full native PAGE for Fig 2d and 3a can be added to the supplementary information.

Please find here the full native PAGE gel of **Fig. 2d**. The lanes that are shown in our manuscript in **Fig. 2d** are framed. As there are, on this same gel, samples that were analysed in conditions not relevant to the study reported in our manuscript, we preferred not to add these gels to the supplementary information section. However, we remain open to discussing this further if the editor and the reviewer feel that it is appropriate.

Please see remark #3 for the native PAGE gel of **Fig3a**.

2. Line 111: two weeks and three months' time...

Can include that the 'population of dimer and tetramer after two weeks and three months' time compared with the population in freshly thawed protein....'

The text was modified to include this suggestion:

(lines 110-111) *“Flash-freezed enriched dimer and tetramer fraction pools were also analysed by mass photometry (MP) at two weeks and three months’ time points after thawing (Fig. 2e).”*

changed to

“Flash-frozen enriched dimer and tetramer fraction pools were also analysed by mass photometry (MP) directly after thawing and at time points of two weeks and three months after thawing (Fig. 2e).”

Color coding in Fig. 2e: the labeling can be in the same order for all peaks (Fresh, 2-weeks and 3 months) It is a little challenging to compare the values in figure.

We agree with the reviewer. Unfortunately, the data were formatted this way at the time of data processing and we no longer have access to the MP software to modify the colours on the histogram in a clean way.

Is the peak height difference due to difference in the concentration of protein used?

Yes, the number of counts (peak height) is directly proportional to the quantity of a protein species in the sample. The mass photometry technique requires extreme dilutions of protein solutions. As the quantity of material was limited, the volumes to be sampled were tiny and it is possible that one sample from the same series was slightly more/less concentrated than the others. Nevertheless, as explained in the manuscript (**Fig. 2d** and **lines 109-110**), we know that total protein concentration (= total number of counts) does not influence the dimer/tetramer ratio and key data in this experiment is the ratio between the number of counts for the peak associated with the dimer and the number of counts for the peak associated with the tetramer.

Also, has the enzymatic activity checked after two weeks and three months, especially for the dimer pool?

The enzymatic activity has not been checked after two weeks and three months in this particular experiment. Nonetheless, we have previously tested samples that had been kept in the fridge for a few days after purification. The specific activity decreases over time but we think this is linked to the quantity of total protein in solution which decreases as a result of precipitation.

The calculated molecular weight by mass photometry is lower than expected tetramer and dimer Mwt, especially for the enzyme after three months. Any degradation or stability issues observed?

The mass photometer had to be calibrated with standard samples of known molecular weight to assign mass to contrast. The molecular weight assigned to each peak of the histogram is therefore relative to the calibration. In view of the non-absolute nature of the molecular weight measurement, and having strongly validated the molecular weight and stoichiometry of the species present in solution by SEC-MALS and SEC-SAXS, we did not take account of the slight difference in the values in relation to the real values. The difference is due to calibration and not to a stability problem.

3. Line 119: Fig. 3a full native page in supplementary information should be added

Please find here the full native PAGE gel. The lanes that are shown in our manuscript in **Fig. 3a** are framed. As there are, on this same gel, samples that were analysed in conditions not relevant

to the study reported in our manuscript, we preferred not to add this gel to the supplementary information section. However, we remain open to discussing this further if the editor and the reviewer feel that it is appropriate.

Nevertheless, since the two lanes are side-by-side, **Fig 3a has been uncropped** to show that the two samples were indeed analysed on the same native PAGE.

4. Line 120: The enzyme kinetic section should be expanded and explained in more detail considering the special nature of this enzyme. If this is not new information a reference for previous studies should be added and stated clearly if these experiments were only repeated for reproducibility.

We thank the reviewer for this remark and **detailed the enzyme kinetics section** as per their recommendations. We made the following changes:

PREVIOUS VERSION - *“The dimeric pool was pure and exhibited uncompetitive substrate inhibition kinetics with a catalytic constant (k_{cat}) of $39.9 \pm 7.8 \text{ s}^{-1}$, a substrate dissociation constant (K_M) of $0.48 \pm 0.17 \text{ mM}$, and a substrate inhibition constant (K_{IS}) of $2.26 \pm 0.70 \text{ mM}$ (Fig. 3b and 3c).”*

REVIEWED VERSION – (lines 123-128) *“The single band on the gel (Fig. 3a, lane 1) demonstrated the purity of the dimeric pool, and the related plot of the dimeric pool activity versus phosphoserine concentration could be fitted to the standard equation for uncompetitive substrate inhibition (equation (1) of the Methods section), with a catalytic constant (k_{cat}) of $39.9 \pm 7.8 \text{ s}^{-1}$, a substrate dissociation constant (K_M) of $0.48 \pm 0.17 \text{ mM}$, and a substrate inhibition constant (K_{IS}) of $2.26 \pm 0.70 \text{ mM}$ (Fig. 3b and 3c). Although a higher K_{IS} value was reported by Grant¹⁹, our results are in agreement with the kinetic behaviour concluded by the author.”*

It is stated as uncompetitive substrate inhibition – are the additional substrate binding sites known? How was this uncompetitive inhibition concluded?

As another reviewer raised the same point, our answer here will overlap in most parts. A velocity curve that rises to a maximum and then descends as the substrate concentration increases is typical of substrate inhibition. The model most commonly used to describe such a phenomenon is that developed by Haldane in 1930. It involves a ternary complex where two molecules of

substrate are bound to the enzyme. The ternary complex either can produce the product at a reduced rate or is unproductive, as is the case here since the initial velocity approaches zero at high substrate concentration (**Figs 3b and 3e**). This substrate inhibition mechanism with an unproductive ternary complex is depicted in **Fig 3c**.

Where the inhibitory substrate molecule binds is not known. Indeed, the formation of the ternary complex can either be interpreted as: (1) the binding of the first substrate molecule at the active site and the binding of the second, inhibitory, substrate molecule at an allosteric site, or (2) the binding of two substrate molecules at the active site in an unproductive manner. Discriminating between the two options in the case of *MtSerB2* and the other mycobacterial PSPs would require further mechanistic studies.

Equation 1 (Methods section) is Haldane's formula for substrate inhibition and is derived from the mechanism of **Fig 3c** using the rapid equilibrium assumption. Substrate inhibition was concluded as **Eq1** fits the data well. The term "uncompetitive" (often overlooked) is specified here to emphasize the fact that the inhibitory substrate molecule, according to the definition of a classical uncompetitive inhibitor, only binds to the enzyme-substrate complex, and not to the free enzyme.

There is clearly inhibition of reaction when using higher substrate concentrations according to Fig. 3b and e. Is this because of substrate binding at a different allosteric site or the product formed being binding at different site? Considering that Ser is also shown to bind allosterically, is the observed inhibition, substrate inhibition or product inhibition? How can that be differentiated?

In the current state of knowledge, we cannot state that the inhibition observed at higher substrate concentrations is not partly due to the presence of L-serine (product) in the reaction solution. We demonstrated that L-Ser inhibits mycobacterial PSPs, indeed binds at allosteric sites at the ACT1-ACT2 interface, and this phenomenon could contribute to the decrease in initial velocity observed at high substrate concentrations. However, we are convinced that *MtSerB2*, *MaSerB* and *MmSerB* do undergo substrate inhibition, whether pure or in conjunction with product inhibition. Indeed, our research led us to study the kinetics of SerB from *Brucella melitensis*. We determined the crystallographic structure of this enzyme (PDB entry: 7QPL), showing that it possesses a single ACT-like domain in N-term unlike mycobacterial SerBs that possess two consecutive ACT domains. As *BmSerB* is monomeric in solution (MALS + SAXS), it does not possess an ACT-ACT interface where L-Ser could bind. Still, this enzyme shows (uncompetitive) substrate inhibition and is left uninhibited by L-Ser, which makes us believe that substrate inhibition occurs at a site distinct from product inhibition in *MtSerB2* and other mycobacterial SerBs.

The inhibiting substrate concentrations are also very high- are they physiologically relevant substrate concentrations?

We agree with the reviewer. As reviewed by Reed *et al.**, substrate inhibition plays a biologically relevant role in the regulation of some enzymes such as tyrosine hydroxylase, acetylcholinesterase, phosphofructokinase, folate cycle enzymes, and DNA methyltransferase. In other cases, as might be for *MtSerB2* given the millimolar inhibition constant, the observed substrate inhibition could be an artifact due to the artificially high substrate concentrations used in the kinetics assays.

*Reed, M. C., Lieb, A., & Nijhout, H. F. The biological significance of substrate inhibition: a mechanism with diverse functions. *Bioessays*. 32(5), 422-429 (2010).

K_{si} – 2.7 mM. Not much sign of inhibition up to 2.0 mM in Fig 8. A

The inhibitory constant K_{is} is the concentration of substrate that is required in order to decrease the maximal velocity of the reaction by half and its value on the v vs $[S]$ plot is hence located after the maximum. According to our kinetic analysis and the fitting of **Equation 1**, the maximal velocity V_{max} was determined to be $55.23 \pm 10.90 \text{ nmol } \mu\text{g}^{-1}\text{min}^{-1}$ (corresponding to a k_{cat} of $39.9 \pm 7.8 \text{ s}^{-1}$). In **Fig 3b**, it can be seen that the velocity curve reaches its maximum at a substrate concentration of 2.0 mM. However, the velocity that is reached by the enzyme at this substrate concentration on the graph is around $30 \text{ nmol } \mu\text{g}^{-1}\text{min}^{-1}$. If the enzyme followed pure Michaelis-Menten kinetics with no substrate inhibition, the graph would have shown a plateau corresponding to the maximal velocity V_{max} ($55.23 \text{ nmol } \mu\text{g}^{-1}\text{min}^{-1}$) from this value of 2.0 mM phosphoserine. The experiment in **Fig 8a** was performed up to phosphoserine concentrations of 2.0 mM to simplify the experimental conditions and study the effect of L-serine on enzyme activity outside the concentration range where the enzyme is strongly inhibited by its substrate. However, although not apparent as such on the graph, inhibition by the substrate is still present at these concentrations. If the experiment had been carried out with concentrations higher than 2.0 mM phosphoserine, the graph would have shown a decrease in the initial rate after 2.0 mM phosphoserine.

Why is the K_M referred as substrate dissociation constant – this is probably true only in some special cases. Is this known for SerB2? If yes, provide appropriate reference. Or it is better to simply mention K_M as Michaelis constant.

All occurrences of “substrate dissociation constant” have been replaced with “Michaelis constant”.

5. Line 126: The dimer percentages slightly different from that in Fig. 2d.

The tetramer fraction pool sample analysed in **Fig 2d** (native PAGE) differs from that measured in **Fig 3d** (MP and kinetics). The sample whose kinetics have been measured contained a larger fraction of dimeric population.

6. Line 143: The domain swapping in MtSerB2 is assumed mainly from homology modeling and fit to SAXS curves. Since the best fitting was achieved after some ‘relaxation’, has the authors also checked non-domain swapped dimers using the same procedure to see if the non-domain swapped model fits well to the SAXS data?

Yes, we also modeled a non-domain swapped dimer from two closed monomers and calculated its scattering curve using CRY SOL. The non-domain swapped dimer model fits the SAXS data equally well than the domain-swapped dimer model constructed by homology modeling after relaxation. This experiment demonstrates that, on the sole basis of SAXS data, it is impossible to determine whether the dimer is domain swapped or not.

We added this important remark in the manuscript text and added new **Fig S5**:

(lines 157-168) *“However, the resolution of structural information provided by SAXS is not sufficient to confirm domain-swapping in MtSerB2. According to domain-swapping principles, a hinge-loop motion is responsible for the opening of a closed MtSerB2 monomer, which further associates with another opened monomer to form the domain-swapped dimer. The opening of the closed monomer is a repositioning of the ACT1 domain and does not involve any refolding. The ACT1-ACT2 intramolecular interactions that are broken upon monomer opening are*

replaced by identical ACT1-ACT2 intermolecular interactions in the dimer. This way, the relative positions of the folded PSP, ACT2, and ACT1 domains in the domain-swapped dimer is strictly identical to the relative positions of these domains in the closed monomer (Supplementary Fig. 5). Hence, a non-domain-swapped MtSerB2 dimer where two closed monomers interact gives a SAXS signature similar to that of the domain-swapped dimer, because the positions of the domains relative to each other are identical and the overall molecular shape is conserved."

Since Ma and MtSerB2 have distinct properties despite their high sequence similarity, it would be good to have additional experimental evidence for the presence of domain swapping in MtSerB2, for example thermal unfolding and/or chemical crosslinking mass spectrometry experiments.

We thank the reviewer for this suggestion. We indeed tried to further investigate the architecture of MtSerB2 oligomers by chemical crosslinking/mass spectrometry, but have not been able to get analysable results through this method. In addition to a challenging experimental setup, regarding the cross-linking conditions that can lead to artifacts, the separation of the cross-linked oligomers by electrophoresis, and the MS method itself, the homomeric nature of the complexes also greatly complicates the identification of cross-linked peptides given the interaction between identical chains. Numerous crystallization trials have been carried out in an attempt to prove domain-swapping in MtSerB2 dimer but unfortunately the enzyme proved resistant to crystallization (please see last point of remark #6).

While it is clear that the studies on MtSerB2 delta Act1 variant confirm that this domain is important for dimer formation, it is not clear how it confirms domain-swapping.

We would like to thank the reviewer for drawing our attention to this important point. Indeed, the link between the monomeric behaviour of the MtSerB2 Δ ACT1 variant and domain-swapping was not clear and an analysis was lacking to support our point. We have **added an analysis of dimeric interfaces** via PDBePISA in the manuscript. The results obtained for the domain-swapped dimer model and a non-domain swapped dimer model were compared and show that the formation of the ACT1-ACT2 interface through ACT1 domain-swapping is crucial to the dimerization of MtSerB2.

Text added:

(lines 168-179) "Therefore, an interface analysis was performed to highlight the importance of forming the intermolecular ACT1-ACT2 interface for MtSerB2 dimerization and to support dimerization by ACT1 domain-swapping.

MtSerB2 domain-swapped dimer homology model and a non-domain-swapped dimer model created in silico from two closed monomers models were submitted to PDBePISA web server. The interface area calculated for the domain-swapped dimer (3025.9 Å²) was about 2.5x larger than the one of the non-domain swapped dimer (1222.3 Å²). The solvation free energy gain (Δ^iG) value computed for the formation of the interface in the domain-swapped model (-31.4 kcal/mol) was 4.4x the one calculated for the non-domain-swapped model (-7.1 kcal/mol). The ACT1-ACT2 intermolecular interface therefore accounts for 60% of the total dimeric interface and more than 75% of the total solvation free energy gain. These values support that the formation of the intermolecular ACT1-ACT2 interface by domain-swapping plays a significant role in stabilising MtSerB2 dimer."

and text added/modified:

(lines 183-185) “From these results and the similar SAXS signatures, we draw the conclusion that the domain-swapped butterfly-like architecture is conserved among MtSerB2, MaSerB, and MmSerB2, and correctly describes all three homodimers in solution.”

There is no mention of any crystallization attempts for this enzyme. Has this been tried and did not crystallize?

Crystallization attempts were made with freshly purified fractions of tetrameric and dimeric MtSerB2. A broad variety of conditions from commercially available screening kits were tried in sitting drop vapour diffusion settings and promising conditions were further optimised. To date, these crystallization attempts unfortunately yielded either salt crystals or non-diffracting crystals of unknown composition. MtSerB2 crystallization could in fact be impaired by the dynamic equilibrium between its quaternary structures.

7. Line 193: monodisperse elution peak – is it not that the tetramer is in equilibrium with some population of dimer?

In native PAGE experiments, we have indeed observed a marginal contamination of the tetrameric population with dimeric population after SEC separation. We think that this is due to the conversion of the tetramer into dimer following separation. The rate of the interconversion might be sufficient for the tetramer to interconvert between separation and native PAGE analysis. However, SEC-UV-SAXS analysis reveals that the peaks obtained are monodisperse. As shown in **Supplementary Fig. 3**, plotting of the R_g across the UV chromatogram results in constant values of R_g at elution times corresponding to the tetramer and dimer peaks. These plateaux of R_g values are indicative that the peaks are monodisperse. The resulting averaged SAXS curves are therefore representative of the tetramer and the dimer, without any sign of cross-contamination.

8. Line 217: The organization of the domains in the final selected model should be explained better. The final selected model from SAXS studies is also seems to be a dimer of dimer (D2 symmetry). To me it looks like that the two dimers are interacting through the C-terminal catalytic (PSP) domain with the ACT domains pointing outwards. Is that correct? i.e similar to model 1 in Fig. 6d but interaction trough the PSB domain. This particular arrangement with the interaction through PSP domain is not depicted in Fig. 6d. It would be good to add this.

We thank the reviewer for drawing our attention to this need for further clarification. We have **reworked Figure 6** to provide more detail on the structure of the final selected tetramer model (see **Fig 6e** in the new version). Along with the cartoon structure, we depicted a schematic representation of the tetrameric arrangement that has also been added to **Fig. 6d** as suggested. Panels have also been rearranged to follow the text better. Changes are as follows (previous version → new version): **Fig. 6e→6f, 6f(left) → 6e, 6f(right) →6g, 6g→6g, 6h→6i, 6i→6j**. We also **added a new supplementary figure (Supplementary Fig. 9)**.

The **following text has been added** to describe new **Fig 6e**:

(lines 245-257) “This tetramer model is non-domain-swapped, as the four monomers are closed to form an intramonomer ACT1-ACT2 interaction. The green monomer is related by C2 rotational symmetry to the yellow, pink, and grey monomers through rotation around three perpendicular C2 axes (respectively C2a, C2b and C2c) hence forming a complex with overall D2 dihedral symmetry. The tetrameric interface is located at the C-terminal catalytic PSP and regulatory ACT2 domains with the ACT1 domains pointing outwards from the complex.

Monomers related through the C2a axis (green-yellow and pink-grey) interact via their PSP and ACT2 domains, while the monomers related through the C2b axis (green-pink and yellow-grey) interact at the ACT2 and ACT1 domains level. It is important to note that the yellow-grey (or green-pink) monomers couple is structurally distinct from the butterfly-like domain-swapped architecture adopted by MtSerB2 dimer in solution: there is here no domain-swapping between monomers and there is a tilt angle between the monomers such that the monomers couple cannot be superimposed on MtSerB2 domain-swapped dimer structure (Supplementary Fig. 9)

Extracts from Fig 6

PREVIOUS VERSION

NEW VERSION

Also, in Fig. 6f, it would be worthwhile to show the relative position of the domains to have better clarity on how the two dimers are interacting and/or color one monomer with different domain colors as in Fig. 1.

Please see the above answer. We addressed this matter accordingly. Fig 6f(left) is now Fig 6e.

Has a similar model with domain swapping also checked for fitting against SAXS data? How would that fit?

We invite the reviewer to **see our answer to the overlapping remark #12**.

The tetramer model is better depicted in Fig. 9, a similar figure can be included in Fig. 6

Please **see our answer to remark #8**. We addressed this matter accordingly. A slight modification has been made to the design of the illustration of the tetramer in Figure 9 to match our new version in Figure 6.

9. Any relation between trimer and tetramer formation?

The inactive tetrameric form of *MtSerB2* has been consistently observed along the active dimeric form at the end of our purification protocol. We believe these two oligomers form spontaneously *in cellulo* after ribosomal translation of *MtSerB2* mRNA. The formation of the inactive trimeric species, however, was only observed *in vitro* when exposing *MtSerB2* dimer to L-ser and, experiments described from **line 275** led us to the conclusion that apparition of the trimeric species could be due to a disruption of the dimeric assembly. To the extent of our knowledge, however, the tetrameric assembly of *MtSerB2* is not affected by the presence of L-Ser, apart from a slight decrease in the L-Ser dose-dependent population, as shown in **Fig.7b**. We therefore believe that the formation of the trimeric species following L-Ser exposition is the mechanism for the natural allosteric inhibition of the phosphatase activity while the spontaneous formation of the tetrameric species may be related to moonlighting properties that are yet to be identified. In line with these distinct purposes for the trimer and the tetramer, we observed by SEC that our *MtSerB2* variant that does not form tetramer (Q92E C148T V149Y G150R I154T) can still form trimer upon exposition to L-Ser and that our *MtSerB2* variant that does not form trimer upon exposition to L-Ser (triple alanine variant, E33A R103A T136A) still shows a tetrameric population (**Fig. 7f**).

10. Is the substrate and product binding allosteric site same? Is the substrate inhibition shown in Fig. 1 different from product inhibition? Fig. 8a shows normal MM kinetic behavior up to 2.0 mM substrate concentration. In Fig. 1 up to 20 mM substrate has been used. Why?

We invite the reviewer to **see our answers to the points 2, 3 and 5 of remark #4** where we provide detailed explanations to all these questions.

11. Line 305: partial (hyperbolic) mixed, predominantly competitive (specific) inhibitor – Explain in more detail. The information is quite scattered now in Figs and methods section. It is important to explain how this conclusion was reached in the results section – from a general reader point of view.

We thank the reviewer for this remark and agree that the original version of our inhibition kinetics section was relatively concise. We followed the reviewer's advice and seize the opportunity to make this section of our manuscript more accessible from the general reader point of view.

As minor clarifications first, we have referred to Baici's book (ref 26) next to the occurrences of the general modifier equation to clarify the origin of the underlying theory, and added a reference to Leskovac's book chapter (ref 82) for a more detailed explanation of the origin of equation 8 and underlying theory of parabolic inhibition for readers who want to go more in detail.

We have **reworked the kinetic inhibition part** of both the **Results** section (from **line 342**) and the **Methods** section (from **line 695**). All modifications and additions are highlighted in red in the text of the revised manuscript. We summarise the changes hereafter:

-We explicitly stated the origin of the mechanism diagnosis methodology used. It is a systematic approach proposed by A. Baici that is based on the general modifier mechanism published in 1953 by Botts and Morales, and that is thoroughly described on the website <https://www.enzyme-modifier.ch>. We are personally convinced that this content is a fantastic and educational tool for any researcher wishing to identify kinetic mechanisms of enzymatic modification unambiguously, systematically and using the appropriate terminology to disseminate the results. We have **added the supplementary table 1** to illustrate our message.

-We have explained and detailed the successive steps of the procedure for processing initial velocities in order to identify the mechanisms.

-In addition the names of the inhibition mechanisms diagnosed for *MtSerB2* on the one hand (partial, predominantly competitive inhibition), and for its homologues *MaSerB* and *MmSerB2* on the other (total parabolic mixed, predominantly uncompetitive inhibition), we further explained their implications at the molecular level, which further highlighted the difference between these mechanisms.

The lettering of the panels in **Fig 8** have been changed: **Fig 8b→8c; Fig 8c→8b**. The legend has been modified consequently and a line about the diagnostic methodology has been added.

Detailed modifications:

Results section, PREVIOUS VERSION : *“from which apparent k_{cat} and KM values could be derived by non-linear regression using the condensed form of the general modifier equation (2). The kinetic mechanism was diagnosed by evaluating the dependences of apparent k_{cat} , KM , and specificity constant (k_{cat}/KM) on L-Ser concentration. Through this method, we identified L-Ser as a partial (hyperbolic) mixed, predominantly competitive (specific) inhibitor of *MtSerB2* (Fig. 8b and 8c), which agrees with Grant’s analysis . The slight partial character was confirmed by the plot of initial velocities versus L-Ser concentration showing a plateau (Fig. 8d).”*

changed to

Results section, NEW VERSION (**lines 342-363**): *“As initial rates decreased with increasing [L-Ser], we could conclude that L-Ser inhibited *MtSerB2* phosphatase activity. To identify the kinetic mechanism of inhibition, we followed the systematic approach proposed by Baici²⁶ (thoroughly described on the website <https://www.enzyme-modifier.ch>). First, apparent k_{cat} and KM values could be determined by fitting the condensed form of the general modifier equation (2) to the v vs $[S]$ data shown in Fig. 8a. Next, the inhibition mechanism was diagnosed by replotting the apparent k_{cat} , KM , specificity constant (k_{cat}/KM), and their respective multiplicative inverses versus [L-Ser] and examining the shapes of the plots (Fig. 8b). As described on Baici’s website, each kinetic mechanism gives a unique combination of replots. The shape of each replot is designated by a letter, depending on how each parameter behaves when [inhibitor] increases (Supplementary Table 1): independent (A, D, H, L, O); increases hyperbolically (B, E, I, M, P); decreases hyperbolically (C, F, J, N, Q) or increase linearly (G, K, R). For *MtSerB2*, the shape of the parameters dependencies on [L-Ser] were found to match the combination C-E-I-N-P (Fig. 8b). This combination corresponds to a hyperbolic mixed, predominantly specific inhibition (‘HMx(Sp>Ca)I’ according to Baici nomenclature) mechanism,*

which, in more common terms, refers to a mechanism of partial, predominantly competitive inhibition. In this mechanism (Fig. 8c), L-Ser can bind to the enzyme-substrate complex but with a lower affinity ($\alpha > 1$) than to the free enzyme (predominantly competitive character). The ternary complex formed between the enzyme, the substrate and L-Ser is still able to form product, but at a slower rate ($\beta < 1$, partial character). This behaviour agrees with Grant's conclusions¹⁹. The partial nature of the inhibition was slightly ambiguous in view of the quasi-linear dependence of apparent $1/k_{cat}$ on [L-Ser] but was confirmed by the plot of initial velocities versus L-Ser concentration showing a plateau rather than reaching zero (Fig. 8d)."

Results section, PREVIOUS VERSION : "These results are in line with our trimer models, whose interfaces formed at the PSP domains could restrict access to the active site."

changed to

Results section, NEW VERSION (lines 367-371): "These results are in line with the formation of a weakly active trimer upon disruption of the dimeric assembly. L-Ser would interact more easily with the free dimer but could also disrupt MtSerB2-phosphoserine complex. Our trimer models, whose interfaces formed at the PSP domains could restrict but not totally prevent access to the active site, explain the remaining low activity."

Results section, PREVIOUS VERSION : "In addition, using the same methodology, we diagnosed L-Ser as a total parabolic mixed, predominantly uncompetitive inhibitor of MaSerB and MmSerB2 (Supplementary Fig. 10)."

changed to

Results section, NEW VERSION (lines 372-379): "In addition, using the same methodology, we diagnosed L-Ser as a total parabolic mixed, predominantly uncompetitive (or 'S-linear I-parabolic noncompetitive') inhibitor of MaSerB and MmSerB2 (Supplementary Fig. 12). In this mechanism, L-Ser can bind both the free enzyme and the enzyme-substrate complex (with more affinity), and a second L-Ser molecule then binds the enzyme-substrate-L-Ser complex to totally inhibit enzyme activity. On the basis of MaSerB-L-Ser cocrystal structures (PDB: 5JLR27 and 5JLP28), we hypothesize that L-Ser would interact with residues D17 and I126 at the ACT1-ACT2 domain interface (Fig. 7d) and that a second binding site leading to total inhibition would appear after a conformational change triggered by substrate binding."

Methods section, PREVIOUS version: "For MtSerB2, the nature of the inhibitory mechanism was determined by assessing the shapes of the 623 dependencies of (...)"

changed to

Methods section, NEW VERSION (lines 695-701): "For MtSerB2, the nature of the inhibitory mechanism was determined according to the methodology proposed by Baici²⁶. This methodology and the associated terminology are thoroughly described on the website <https://www.enzyme-modifier.ch>, where is offered a precise and systematic approach to the diagnosis of the kinetic mechanisms of enzymatic inhibition or activation by a modifier compound. As per the approach described above, we were able to identify the nature of the

kinetic mechanism of MtSerB2 inhibition by L-Ser by examining the shapes of the plots showing the respective dependencies of (...)

Effect of phosphate ion for the inhibition not mentioned – discussed in Grant (Biochemistry).

To the best of our understanding, in Biochemistry, G.A. Grant states that product (L-Ser) inhibition happens only in the presence of phosphate ions in *Mycobacterium tuberculosis* phosphoglycerate dehydrogenase (*MtPGDH*), the first enzyme of the L-Ser biosynthetic pathway (see the figure below). Given that *MtPGDH* can only be inhibited by L-Ser in the presence of 30-50 mM phosphate (previous study), Grant hypothesises that L-Ser feedback inhibition of *MtSerB2* ('*mtPSP*' in their work) acts as a secondary control point in the L-Ser biosynthetic pathway when phosphate concentrations are not high enough for *MtPGDH* to be feedback inhibited. In *MtSerB2*, inhibition will always happen in presence of phosphate ions, as phosphate is a product of the dephosphorylation reaction (phosphoserine to serine) it catalyses and also a classical dead-end competitive inhibitor of the enzyme, as per Grant's conclusions.

12. Can the SAXS data differentiate between domain-swapped and closed dimer/tetramer? – probably not. Whether it is closed or domain-swapped dimer/tetramer, the interaction between Act1 and Act2 are the same. Therefore, for the final best fitting models, the corresponding domain-swapped or closed oligomer fitting should also be tested.

As we clarified in the text (lines 157-168) following remark #6, the resolution of structural information provided by SAXS is indeed not sufficient to differentiate between a domain-swapped and a closed oligomer. In this paragraph, we explained that a non-domain-swapped dimer fits the SAXS data equally well, but that ACT1 domain-swapping is supported by the results of the interface analysis, the monomeric behaviour of variant *MtSerB2* Δ ACT1 and the high similarity between *MaSerB* and *MtSerB2* SAXS curves. For the tetramer, like we replied to another reviewer making a similar comment, we present converging observations throughout the manuscript that support the closed model obtained via SAXS-based screening. These are the following:

First, as shown in Fig. 2d and discussed at lines 109-110 and 241-243, we draw the reader's attention to the observation that concentrating the dimeric fraction up to 25.5 mg/mL does not result in the formation of the tetramer. If the tetramer was a dimer of domain-swapped dimers, we would expect that such a high concentration favours interactions between dimers and hence forces tetramer appearance in solution. However, it is not the case. Conversely, the conversion of a domain-swapped dimer to a tetramer made of four closed monomers involves a dissociation step of the dimer into its monomeric constituents. In such a set-up, concentrating the dimer would not result in formation of the tetramer. That is what is observed.

Secondly, the 'four closed monomers' architecture accounts for the observations we made regarding the relative stabilities of the tetramer and the dimer, and their rate of interconversion.

As highlighted by mass photometry (**Fig 2e**), the tetramer quickly converts to dimer while the dimer needs three months to reform a population of tetramer. As discussed **from line 411** and depicted in **Fig 9e**, this is probably because the separation of four stable closed monomers (that are in dynamic equilibrium with their open form according to domain-swapping principles - see ref 33) is less energetically costly than the separation of a more stable intertwined dimer into two open monomers of high energy.

Finally, the ‘four closed monomers’ architecture can also explain why *MtSerB2* forms a small population of tetramer in addition to the more stable dominant dimeric population: based on **Fig 9e**, right after ribosomal translation, the freshly translated monomeric chain would probably adopt its most stable conformation: a closed conformation (with ACT1-ACT2 intramolecular interactions). We discuss this **from line 416**. The formation of the dimer is favoured by thermodynamic factors: it is the most stable oligomer but its formation requires overcoming an energy barrier imposed by the opening of the monomer. Conversely, the formation of the tetramer (less stable than the dimer, but more than an isolated monomer) would be kinetically favoured, as this process only needs four monomers to interact together without energetically costly conformation changes.

Additionally, in the domain-swapped homology model of *MtSerB2* dimer, the hinge loop adopts an extended conformation to link R83 to H96, whose C α are separated by 29.9Å. In the tetramer model that we propose, the distance between R83 and H96 of adjacent subunits is 39.9Å. In that configuration, it would require the hinge-loop to be completely extended to connect these two anchoring points, which would be strongly entropically unfavourable.

Although we understand that higher resolution structural techniques would be needed to univocally validate our model, these observations converge and support that *MtSerB2* tetramer is most likely composed of four closed monomers.

13. Line 871 Fig. 1 legend: ACT domains not yellow and purple but yellow and red

“Purple” changed to “red”.

14. Line 872 Fig. 1 legend: The Mg²⁺ ion in green is not visible. The color of Mg²⁺ ions should be changed or highlighted for better visibility.

The color of Mg²⁺ ion **has been changed** to pink (same as the Mg²⁺ ion of the left subunit on **Fig. 1**) for better visibility and the **legend of Fig. 1 has been modified** accordingly. Although it is true that the ion on the right is partly hidden by the end of a Rossmannoid fold helix, the position of both ions is identical in each subunit. The position of the right-hand ion can therefore be deduced from the symmetry with the left-hand ion.

Reviewer #3 (Remarks to the Author):

Pierson et al provide outstanding evidence that SerB2 from *M. tuberculosis* uses the morpheein model of allostery to control catalysis and that pharmacological stabilization of one or more alternate assemblies is a promising approach to new therapies. The work combines a variety of methods, both experimental and computational, directed mutagenesis, as well as sophisticated modelling to fit SAXS data, to support the existence of a flexible monomer, a dimer, at least one tetrameric assembly, and

an unexpected trimeric assembly that is the L-Ser-stabilized allosterically inhibited form. Suggestion that alternate assemblies may correlate with SerB2 moonlighting functions further defines the morpheein character of SerB2. So, why does the title use the term morpheein-like? Strongly suggest removing the “-like” from the title. SerB2 uses the morpheein model of allostery.

Author response:

We would like to thank Prof. E.K. Jaffe for the very positive assessment of our manuscript and her confidence regarding our findings, which is highly encouraging. We originally used the term “morpheein-like” to describe the oligomeric properties of *MtSerB2* as we were aware of the remark below. We have modified all occurrences of “morpheein-like” to “morpheein” in the title and the text according to the suggestion.

The only significant concern is the use of the term induce. The morpheein model of allostery is not an induced fit model. The ligand does not bind to one form and induce a conformational change. The protein exists in an equilibrium of assemblies wherein the ligand binding site is on one assembly and not on the others. The authors say the different assemblies coexist, interconverting through a conformationally flexible monomer. The mechanism is conformational selection. Note however the description lines 283-286. Does L-Ser bind to the dimer? Or, does L-Ser bind to the trimer and pull the equilibrium in that direction? See also lines 361-365 and lines 1043-1044.

We thank E.K. Jaffe for raising this interesting concern which allows us to open the discussion on the concept of the morpheein model of allostery in our manuscript.

In our work, we present clues that the first step of the oligomeric transition is a disruption of the dimeric species. Specifically, modelling and docking work reported at **Figures 7d** and **7e** and discussed at lines 306-315 unveiled an L-ser interacting pocket specific to *MtSerB2* dimer located at the ACT1-ACT2 interface. This pocket is supposedly absent in the trimer according to SAXS modelling (**Fig 7g**). Also, when mutating the identified L-Ser interacting residues to alanine (**Fig 7f, lines 315-322**), the dimeric species no longer undergoes a transition to the trimeric species. Whether L-Ser could remain bonded to the open monomer (and hence to the trimer) after disruption is a reasonable possibility but it remains hypothetical at this stage.

However, it is clear that a transient monomer able to populate alternate conformations allows interconversion between the active dimer to the inactive trimer - and tetramer. This is one of the distinctive features of the proteins called morpheeins. Nevertheless, we are well aware that the morpheein model of allostery is, as stated by E.K Jaffe, defined by a mechanism of conformational selection and not a mechanism of disruption. This is why, we had originally titled our article “A morpheein-like equilibrium [...]”.

We propose to adapt our discussion (**lines 443-447**) to highlight this subtlety and speculate that the morpheein model of allostery could be extended to include allostery by oligomeric disruption under its umbrella. In the present situation, we have also chosen to keep the term “induced”, which is directly linked to our observation that adding L-Ser to the solution causes a dimer-to-trimer shift by disrupting the dimer structure.

New paragraph added to the discussion:

(lines 443-447) *“However, the morpheein model of allostery is originally defined by a mechanism of conformational selection in which a stabilising ligand binds to one assembly, not the others, and displace the equilibrium towards this assembly. It does not imply a ligand-induced disruption of one assembly. Our results therefore suggest that the definition of*

morpheins could potentially be extended to include allostery by oligomeric disruption and hence encompass a larger repertoire of enzymes.”

Other minor modification to keep in line with the added paragraph:

(lines 449-451): “...as the enzyme PSP activity could be inhibited by compounds designed to stabilise the (nearly) inactive trimeric and tetrameric forms.”

changed to

“...as the enzyme PSP activity could be inhibited by compounds designed to disrupt the active dimer or stabilise the (nearly) inactive trimeric and tetrameric forms.”

Minor points, suggestions, and corrections:

1. The authors might point out that the conformational change in the monomer (open vs closed) is a repositioning of folded domains, a hinge motion, and does not include any refolding (as is found in metamorphic proteins).

Additions were made to the text to explain this feature of domain-swapping in more detail :

(lines 158-161): “According to domain-swapping principles, a hinge-loop motion is responsible for the opening of a closed MtSerB2 monomer, which further associates with another opened monomer to form the domain-swapped dimer. The opening of the closed monomer is a repositioning of the ACT1 domain and does not involve any refolding.”

We also agree that the term “folded monomer” was confusing. The word “folded (on itself/themselves)” **was replaced by** “closed” when referring to the monomer in the text.

2. The authors might point out that control of catalysis depends on quaternary structure changes that control active site access. This is a common theme among morpheins.

We **added two sentences** to clarify this point and argument:

(lines 432-434) “The distinct activity levels of the oligomers could be structurally interpreted by facilitated access to active site in the dimer and hindered access in the trimer and tetramer.”

(lines 441-442) “The control of catalysis is structurally explained by a quaternary structure change that modulates active site access^{36,37}.”

3. Proteins with amino acid substitutions should be called variants. They are not mutants (e.g. bottom of p. 7).

All occurrences of the word “mutant” have been **replaced by** “variant”

4. Since it takes only a few specific amino acid substitutions to cause SerB2 to lose the ability to populate alternate assemblies, how might this relate to the eventual development of resistance to the promised therapeutic?

We indeed showed that only a few mutations would be needed for *MtSerB2* to remain dimeric and lose its morphein properties. However, *MtSerB2* close orthologs *MaSerB* and *MmSerB2* exhibit a distinctive mechanism of allosteric regulation (**Fig 8 vs Supplementary Fig 12, lines 372-382**) likely involving signal transduction from the allosteric site to the active site. Because *MtSerB2* appears to have naturally evolved a morphein mechanism to control its activity, it is unlikely that the enzyme also possesses the same allosteric path as its close orthologs *MaSerB* and *MmSerB2*. Hence, losing the morphein properties would most likely result in an

unregulated *MtSerB2*, unable to detect and respond to an abundant presence of L-Ser. This would result in the loss of an important regulation check-point of the L-serine biosynthesis in *Mtb* and might lead to the progressive accumulation of L-serine up to toxic levels. In that way we believe that the emergence of resistance to a therapeutic targeting the morphoein behaviour of *MtSerB2* could be delayed.

5. Fig 1a – legend – the N-terminal ACT domains are not yellow and purple. They are yellow and red. Reviewer cannot see the green sphere.

“Purple” **changed** to “red”. The colour of Mg^{2+} ion has been **changed** to pink (same as the Mg^{2+} ion of the left subunit on **Fig. 1**) for better visibility and the legend of **Fig. 1** has been **modified** accordingly. Although it is true that the ion on the right is partly hidden by the end of a Rossmannoid fold helix, the position of both ions is identical in each subunit. The position of the right-hand ion can therefore be deduced from the symmetry with the left-hand ion.

6. Fig 2d – lanes 1 and 2 suggest the presence of two conformationally distinct tetramers. Although it is not possible to investigate this using SEC, one may be able to separate these forms using IEC. This is a suggestion for future study.

It is indeed unclear in **Fig. 2d** whether there might be one or more conformationally distinct tetramers due to a slight smear of the upper bands. However, SEC-UV-SAXS analysis reveals that the tetrameric peak is monodisperse. As shown in **Supplementary Fig. 3**, plotting of the R_g across the UV chromatogram results in constant values of R_g at elution times corresponding to the tetramer and dimer peaks. These plateaux of R_g values are indicative that the peaks are monodisperse. We would therefore argue that the slightly smeared tetrameric band in **Fig. 2d** is rather attributable to gel electrophoresis effects. This effect was not observed in independent native PAGE experiments, as exemplified in **Fig. 3a**.

7. Line 887 – change to “at a 90° angle”.

Changed.

8. Note that Fig 3a shows a clean single tetramer band. How is the sample different from Fig 2d?

Although not from the same purification batch, the samples were prepared similarly. The differences in the aspect of the gels are most likely attributable to electrophoresis effects - like the overheating of the electrophoretic cassette - and not to a relevant enzyme behaviour. In **Fig. 2d**, the dimeric band is also affected in a comparable manner, although less pronounced.

9. Line 901 – replace flash-freezed with flash-frozen.

All “flash-freezed” occurrences have been **replaced** with “flash-frozen”.

10. Line 950 – what is a low-conversation score?

It is a typo. “low conversation” has been **replaced** with “low conservation”.

11. Where possible, make the text in the figures larger.

The size of the text in the figures **has been increased**.

12. Fig 7d requires higher quality images.

The quality of the images provided with the version intended for reviewing was indeed not sufficient. This **issue is solved** in the final high-quality image.

13. Fig 8 – Remove the artifact along the right edge of the figure.

The artifact probably appeared when converting the document and adding line numbers. This problem will be solved when uploading the figure files and processing the final version of the manuscript.

E.K. Jaffe

REVIEWERS' COMMENTS:

Reviewer #1 (Remarks to the Author):

In this revised manuscript, the authors have adequately addressed all of my concerns and comments. There is only one thing that I would suggest omitting. On line 78, the authors use the term "eye opening". While the manuscript may present evidence for the unexpected, the use of this wording seems a bit exaggerated to the point of seeming self-praising. Perhaps it might be revised to something a bit less exclamatory.

Reviewer #2 (Remarks to the Author):

The revised manuscript has adequately addressed most of the points raised during the first round of review. The responses to the review questions are well justified. Appreciate the authors for carefully revising the manuscript. I recommend the acceptance of the manuscript in this revised form.

REVIEWERS' COMMENTS:

Reviewer #1 (Remarks to the Author):

In this revised manuscript, the authors have adequately addressed all of my concerns and comments. There is only one thing that I would suggest omitting. On line 78, the authors use the term "eye opening". While the manuscript may present evidence for the unexpected, the use of this wording seems a bit exaggerated to the point of seeming self-praising. Perhaps it might be revised to something a bit less exclamatory.

Author response: We thank the reviewer for the comment and for taking the time to review our revised manuscript. As suggested, we removed the term "eye opening" on line 78 in the revised version.

Reviewer #2 (Remarks to the Author):

The revised manuscript has adequately addressed most of the points raised during the first round of review. The responses to the review questions are well justified. Appreciate the authors for carefully revising the manuscript. I recommend the acceptance of the manuscript in this revised form.

Author response: We thank the reviewer for the positive feedback on our revisions.